# Probabilistic forecasting of monthly dengue cases using epidemiological and climate signals: A BiLSTM-Negative Binomial Model versus Mechanistic and Count-Model Baselines

Michael Marko Sesay[1]*, Antony Ngunyi[2], Herbert Imboga[3]

**1** Department of Mathematics, Pan African University Institute for Basic Sciences, Technology and Innovation, Kiambu, Kenya, **2** Department of Statistics and Actuarial Sciences, Dedan Kimathi University of Technology, Nyeri, Kenya, **3** Department of Statistics and Actuarial Sciences, Jomo Kenyatta University of Agriculture and Technology, Kiambu, Kenya

* sesay.michael@students.jkuat.ac.ke

## Abstract

Reliable short-term forecasts enable urban health systems to anticipate dengue surges and allocate resources effectively. We assembled monthly dengue case counts for Freetown, Sierra Leone (2015–2024), and compared four probabilistic model families under a leakage-safe, rolling-origin evaluation at 1–3-month horizons: a negative binomial generalized linear model (NB-GLM), a negative binomial ING-ARCH model (INGARCH-NB), a mechanistic renewal model with negative binomial observations (Renewal-NB), and a bidirectional long short-term memory network with a negative binomial output (BiLSTM-NB). All models used the same seasonal harmonics and autoregressive lags; "light" climate inputs (rainfall, temperature, and relative humidity) were restricted to lag-1 covariates to reflect real-time availability. We evaluated probabilistic performance using mean log score (primary), empirical coverage, and median widths of 50% and 90% predictive intervals, calibration diagnostics based on the probability integral transform, and Diebold-Mariano tests with Newey-West standard errors. For the main comparison, we evaluated models on a strictly matched set of common issue-target pairs within each horizon ($n = 32$ per horizon). On this aligned set, INGARCH-NB achieved the best mean log score at all horizons, indicating the strongest overall distributional accuracy. BiLSTM-NB remained competitive and provided more conservative upper-tail uncertainty at longer horizons (e.g., 90% interval coverage of 100% at $h = 3$), at the cost of wider intervals. NB-GLM variants produced the sharpest intervals but were substantially undercovered, indicating overconfidence, while renewal-based forecasts attained nominal coverage largely through uncertainty inflation that degraded sharpness and log score. In a leakage-safe light-climate ablation, adding lag-1 climate covariates yielded small, statistically non-significant gains for NB-GLM and did not improve renewal forecasts. Overall, the results support a horizon-aware toolkit for operational dengue

**Data availability statement:** De-identified monthly dengue cases and climate aggregates (2015–2024) are publicly available at https://doi.org/10.34740/kaggle/dsv/13257213.

**Funding:** The author(s) received no specific funding for this work.

**Competing interests:** The authors have declared that no competing interests exist.

forecasting: INGARCH-NB as a strong default when distributional accuracy is prioritized, complemented by calibrated deep learning (BiLSTM-NB) when conservative tail reliability is preferred. The aligned indices, per-issue forecasts, and code provide a transparent baseline for future work in similar urban settings.

---

## Author summary

We conducted this study to help public health teams in Freetown, Sierra Leone, plan clinical capacity and vector control using reliable short-term dengue forecasts. Many forecasting approaches exist, but they are rarely compared under the same leakage-safe conditions. We assembled monthly dengue case data (2015–2024) and built consistent seasonal and autoregressive predictors for all models, using only a lightweight set of lagged climate inputs feasible in real time. We compared four model families: a negative binomial generalized linear model, an INGARCH count model, a mechanistic renewal model, and a bidirectional LSTM with a negative binomial output. Using an expanding-window, rolling-origin evaluation at 1- to 3-month horizons, we assessed probabilistic accuracy with proper scoring rules, predictive interval coverage, probability integral transform (PIT) histograms, and Diebold–Mariano tests on aligned targets. No single method dominated across all horizons: parsimonious count models performed best at 1–2 months, while a calibrated BiLSTM excelled at the 3-month horizon and provided reliable uncertainty estimates. These findings support a horizon-specific toolkit for operational dengue forecasting in similar urban settings.

## Introduction

Dengue fever remains one of the most pervasive vector-borne diseases worldwide, affecting tropical and subtropical regions with an expanding geographic footprint. The spread of dengue continues to accelerate, driven by recurring outbreaks that cause substantial morbidity and strain public health systems. This expansion is fueled by complex interactions between climatic conditions, which shape mosquito breeding habitats and virus survival, and rapid urbanization, which increases human–mosquito contact. These factors collectively amplify transmission dynamics, triggering more frequent outbreaks and posing persistent challenges for disease control. Given these complexities, there is a critical operational need for reliable short-term forecasting tools to enable health authorities to anticipate dengue incidence and allocate resources efficiently [1–3].

Among regions facing emerging dengue risks, West Africa presents epidemiological and surveillance characteristics that warrant focused attention. Accumulating evidence highlights sustained local dengue transmission, contradicting earlier assumptions that cases are primarily sporadic or imported. The region's ecological and socio-economic context—including seasonal rainfall patterns, temperature

variability, and rapid urban growth—influences mosquito population dynamics and dengue transmission potential. This evolving landscape underscores the need for improved situational awareness through enhanced surveillance and data-driven forecasting. Timely, region-specific information is crucial for mobilizing interventions and containing outbreaks that impose considerable health and economic burdens on vulnerable populations [4–6].

Freetown, the capital of Sierra Leone, serves as a pertinent setting for operational dengue forecasting given the availability of routine surveillance data at a monthly cadence and documented dengue activity. Its coastal urban environment and climatic conditions support vector proliferation, creating a practical need for forecasting to inform public health decision-making. Monthly forecasting represents a pragmatic compromise between data availability and operational utility: it aligns with common reporting workflows and supports planning for staffing, diagnostics, and vector-control activities on a near-term horizon. However, because monthly aggregation can obscure rapid shifts in incidence, we emphasize leakage-safe evaluation and robustness checks when comparing model classes at this time scale [7,8].

Monthly dengue counts typically exhibit overdispersion, strong annual seasonality, short serial dependence, and potentially non-linear relationships with environmental drivers, posing challenges for standard time-series approaches [9,10]. Negative binomial generalized linear models (NB-GLMs) offer interpretable covariate effects and handle overdispersion, but they may inadequately represent temporal feedback dynamics [11,12]. Negative binomial INGARCH models explicitly incorporate the dependence of the conditional mean on both past observations and past conditional means, providing an observation-driven approach for count time series [13,14]. Renewal models link incidence to a time-varying reproduction number ($R_t$) and a serial-interval kernel, supporting epidemiological interpretation while remaining parsimonious [15]. Modern bidirectional long short-term memory (BiLSTM) architectures with negative binomial output heads can learn non-linear patterns while producing probabilistic count forecasts [16]. However, time-series machine learning remains susceptible to information leakage through improper feature construction and validation design, necessitating careful feature timing and rolling-origin evaluation.

Despite extensive methodological development, leakage-safe and aligned comparisons of regression baselines, observation-driven count models, mechanistic renewal formulations, and deep sequence models for monthly dengue forecasting in West Africa remain limited [17]. We address this gap through an aligned, expanding-window evaluation in Freetown comparing four models: an NB-GLM, INGARCH-NB, Renewal-NB, and a BiLSTM-NB architecture featuring autoregressive skip connections and optional isotonic calibration. We analyze monthly reported dengue cases in Freetown (2015–2024) alongside monthly rainfall, air temperature, and relative humidity aggregates as potential environmental drivers. To reflect operational feasibility, we utilize a "light" climate feature set—limited to three variables—and apply conservative lagging rules; we also report sensitivity analyses examining alternative climate specifications and key mechanistic assumptions. All models incorporate 12-month harmonic terms and autoregressive lags ($y_{t-1}, y_{t-2}, y_{t-3}, y_{t-12}$) to capture seasonal and short-term dependence, and we enforce leakage-safe timing for all inputs.

We employ an expanding-window rolling origin protocol for forecast horizons of $h \in 1, 2, 3$ months, using a minimum training length of 48 months to stabilize seasonal estimation. Evaluation prioritizes the mean log score as a strictly proper scoring rule, alongside 50% and 90% prediction-interval coverage and median interval widths, to summarize calibration and sharpness. Distributional calibration is assessed using probability integral transform (PIT) diagnostics adapted for counts, while the statistical significance of forecast differences is tested using Diebold–Mariano tests with Newey–West standard errors on aligned issue–target indices [18]. Our contributions include:

- Leakage-safe feature timing, including conservative lagging of climate inputs; a seed-ensemble BiLSTM-NB with autoregressive skip connections; and optional isotonic calibration to improve reliability

- A head-to-head comparison of NB-GLM (direct forecasting), INGARCH-NB (observation-driven), Renewal-NB (mechanistic), and BiLSTM-NB under shared seasonal and autoregressive structure

- Aligned backtesting enabling fair Diebold–Mariano comparisons, with unaligned results preserved in supplementary materials

- Operational evaluation emphasizing proper scoring rules, reliability, and sharpness for public health decision support, alongside robustness checks for key modeling assumptions

The remainder of this paper is organized as follows: Section 2 describes data sources, feature engineering, model formulations, experimental setup, and evaluation metrics. Section 3 presents comparative results, diagnostic analyses, and robustness checks. Section 4 discusses implications for operational dengue forecasting in resource-limited settings and identifies future research directions.

## Materials and methods

### Study setting, outcome, and covariates

**Study setting and time span.** We curated a dengue surveillance and climate dataset for Freetown, Sierra Leone, spanning January 2015 to December 2024. The dataset links monthly dengue case totals to monthly meteorological summaries to support leakage-safe probabilistic forecasting at 1–3 month horizons (see S1 Data).

**Dengue surveillance outcome.** Let $Y_t \in \mathbb{N}_0$ denote the number of reported dengue cases in month $t$. Each observation represents the *total* count of laboratory-confirmed and clinically suspected dengue infections recorded in the Freetown catchment during that calendar month.

**Climate covariates.** Monthly climate covariates were obtained from publicly available meteorological sources and aligned to the dengue reporting calendar: precipitation (mm; monthly total), near-surface air temperature (°C; monthly mean), and relative humidity (%; monthly mean). These covariates were selected because they are plausibly linked to *Aedes* mosquito ecology and dengue transmission and because they are readily available in operational settings.

**Exploratory summary.** To characterize seasonality and interannual variability at the monthly scale, we summarize the dengue series using (i) a time plot and (ii) an average monthly profile ( Figs 1 and 2). The series exhibits pronounced annual seasonality with recurrent peaks, motivating the inclusion of seasonal harmonic terms and autoregressive lags shared across all model classes.

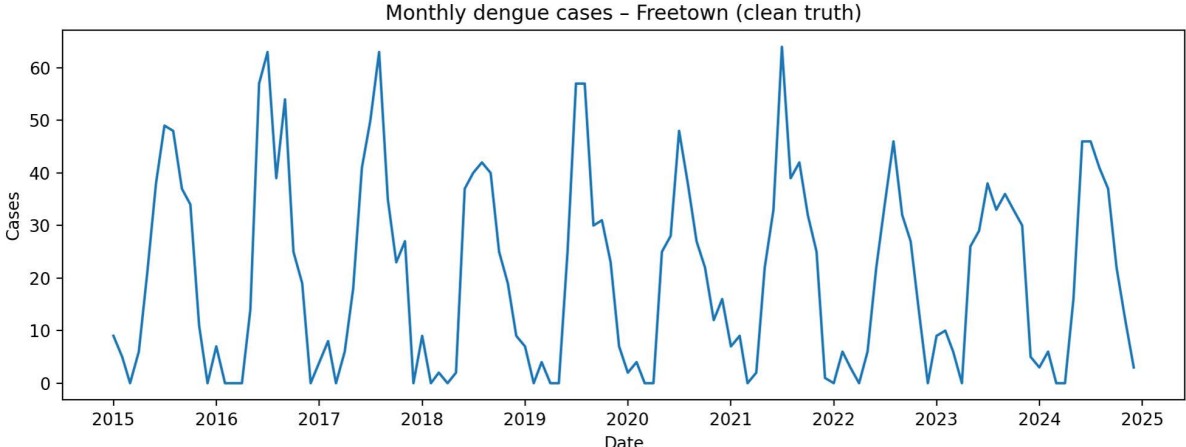

**Fig 1. Monthly reported dengue cases over the study period, showing clear seasonality and recurrent annual peaks in Freetown.**

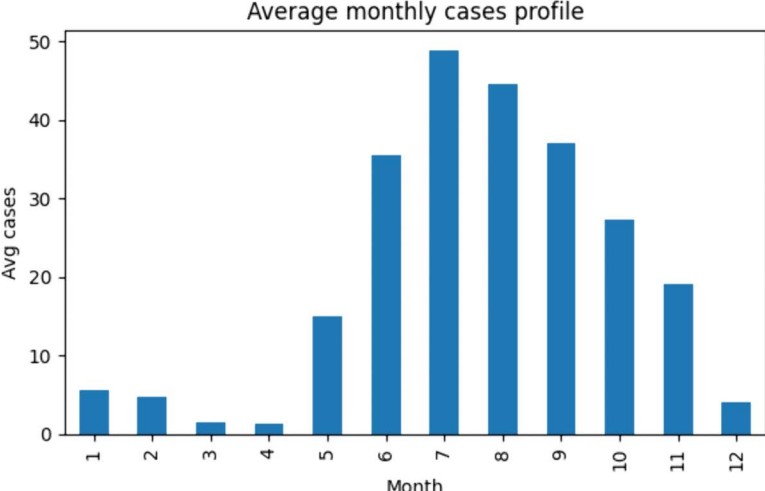

**Fig 2. Average monthly case profile highlighting the seasonal transmission pattern in Freetown.**

## Preprocessing and feature engineering

**Calendar alignment and outcome construction.** All records were aligned to a complete monthly calendar from January 2015 to December 2024. The analysis outcome is the monthly count $Y_t$ in month $t$, with $Y_t \geq 0$ by construction.

**Seasonal harmonics and autoregressive lags.** To represent annual seasonality, we construct 12-month trigonometric harmonics from the calendar month index $m_t \in \{1, \ldots, 12\}$:

$$s_t = \sin\left(\frac{2\pi m_t}{12}\right), \qquad c_t = \cos\left(\frac{2\pi m_t}{12}\right). \tag{1}$$

Short- and medium-range dependence is represented by integer lags of the case series.

$$\left\{ Y_{t-1}, \ Y_{t-2}, \ Y_{t-3}, \ Y_{t-12} \right\}. \tag{2}$$

Lags are used only when available; issue months without the required lagged values (e.g., at the beginning of the series) are excluded for the relevant model/horizon.

## Climate feature set and leakage-safe timing

Environmental drivers are limited to a "light" set of at most three monthly climate aggregates, precipitation, temperature, and relative humidity, to reflect operational feasibility. To prevent look-ahead, climate covariates used for forecasting issue month $t$ are restricted to values available at or before $t$. In the primary specification, we adopt conservative lagging:

- For all models, precipitation is used at lag-1 by design.

- For NB-GLM, INGARCH-NB, and Renewal-NB, temperature and humidity are used at lag-1 in the primary analysis. Contemporaneous values are considered only in a sensitivity analysis under an explicit assumption about reporting latency.

- For BiLSTM-NB, *all* climate inputs are strictly lagged by one month: $X_{t-1}^{(rain)}$, $X_{t-1}^{(temp)}$, $X_{t-1}^{(rh)}$.

Let $x_t$ denote the vector of selected climate covariates at month $t$. Climate covariates are standardized within each training fold by

$$\tilde{x}_t = \operatorname{diag}(\hat{\sigma}^{-1})(x_t - \hat{\mu}), \tag{3}$$

where $\hat{\mu}$ and $\hat{\sigma}$ are computed *only* on the current training window and applied to the corresponding validation/test issues. Trigonometric harmonics $(s_t, c_t)$ are scaled analogously. Count lags $\{Y_{t-\ell}\}$ are left unscaled.

**Recurrent-model inputs and targets.** For recurrent models, we form leakage-safe supervised sequences with a fixed lookback $W = 12$ months. The count stream for issue $t$ is

$$\mathbf{C}_t = (Y_{t-W+1}, Y_{t-W+2}, \ldots, Y_t)^\top \in \mathbb{R}^{W \times 1}, \tag{4}$$

and the auxiliary feature vector is

$$A_t = \left[ s_t, c_t, Y_{t-1}, Y_{t-2}, Y_{t-3}, Y_{t-12}, \tilde{x}_{t-1}^\top \right]^\top, \tag{5}$$

with climate at lag -1 only. We construct multi-step targets for $h = 1,2,3$ months ahead,

$$\mathbf{Y}_t^{(\text{tar})} = (Y_{t+1}, Y_{t+2}, Y_{t+3}). \tag{6}$$

Only issues for which all required elements of $\mathbf{C}_t$ and $A_t$ are present are used for training and evaluation. Targets $Y_{t+h}$ are never used in feature computation at issue $t$.

**Audit trail for alignment.** To support auditability, we persist per-issue (`issue_date`, `target_date`) keys and fold-specific scaling statistics used at each issue. These artifacts allow exact regeneration of aligned evaluation sets and verification of leakage safeguards.

**Missing data handling.** After alignment to a complete monthly calendar (January 2015-December 2024; $T = 120$ months), we verified the completeness of the dengue outcome and the selected "light" climate covariates (precipitation, temperature, and relative humidity). The aligned analysis table contains no missing values in $Y_t$ or in any selected climate variable (0/120 missing months for each field; 0.0% missing overall). Consequently, **no imputation was performed** and **no months were dropped due to missingness** in the primary analysis.

**Sensitivity (not applicable for this dataset).** Because there are no missing values in the aligned series, missing-data sensitivity analyses (e.g., alternative imputation strategies or complete-case versus imputed comparisons) are not applicable. We state this explicitly to document that the absence of imputation reflects data completeness rather than an omitted methodological detail.

**Operational procedure under missingness (deployment guidance).** In prospective operational settings where climate feeds may be delayed or incomplete, missing covariates should be handled *within each rolling-origin training fold* to preserve leakage safety (i.e., imputation parameters computed using training data only, then applied to the corresponding forecast issue). Missing dengue outcomes should not be imputed as forecast targets; instead, affected issue-target pairs should be excluded from scoring and clearly logged in the per-issue audit trail (issue date, target date, and missingness flags).

## Models

All models produce probabilistic forecasts for monthly counts using a negative binomial (NB2) observation model with mean $\mu$ and dispersion $\alpha$, where $\operatorname{Var}(Y \mid \mu) = \mu + \alpha\mu^2$. Except where noted, we fit separate direct models per horizon ($h$).

Model inputs follow the leakage-safe timing rules in Sections miss-preprocess, and all standardization parameters are estimated on training folds only.

## Probabilistic forecasting models

All approaches produce probabilistic forecasts for monthly counts under a negative-binomial NB2 observation model with mean $\mu$ and dispersion $\alpha$, such that $\text{Var}(Y \mid \mu) = \mu + \alpha\mu^2$. Unless otherwise stated, models are trained and evaluated separately for each forecast horizon $h \in \{1, 2, 3\}$ using a direct strategy. Model inputs follow the leakage-safe timing rules in Sections preprocess and climate_timing, and all standardization parameters are estimated on training folds only. For likelihood computations, we use the $(r,p)$ parameterization with $r = \alpha^{-1}$ and $p = r/(r + \mu)$.

## NB-GLM

The negative binomial generalized linear model (NB-GLM) extends the Poisson GLM to accommodate overdispersion commonly observed in dengue counts [11,12]. We adopt the NB2 mean-variance relationship. For horizon $h \in \{1, 2, 3\}$, the monthly count $Y_{t+h}$ conditional on the information set $\mathcal{F}_t$ is modeled as

$$Y_{t+h} \mid \mathcal{F}_t \sim \text{NB2}\left(\mu_{t,h}, \alpha\right), \qquad \text{Var}(Y_{t+h} \mid \mu_{t,h}) = \mu_{t,h} + \alpha\,\mu_{t,h}^2, \tag{7}$$

where $\mu_{t,h} = \mathbb{E}[Y_{t+h} \mid \mathcal{F}_t]$ and $\alpha > 0$ is the overdispersion parameter [19]. As $\alpha \to 0$ the model approaches Poisson, larger $\alpha$ implies greater overdispersion [20]. For likelihood computations we use the $(r,p)$ parameterization

$$r = \alpha^{-1}, \qquad p = \frac{r}{r + \mu_{t,h}}, \tag{8}$$

with pmf

$$\Pr(Y_{t+h} = y \mid \mu_{t,h}, \alpha) = \frac{\Gamma(y + r)}{\Gamma(r)\,\Gamma(y + 1)}\, p^r\, (1 - p)^y, \quad y = 0, 1, 2, \ldots \tag{9}$$

We use a horizon-specific linear predictor (direct strategy) [21]:

$$\log \mu_{t,h} = \mathbf{x}_t^\top \beta_h, \qquad \mathbf{x}_t = \left[1,\ s_t,\ c_t,\ Y_{t-1}, Y_{t-2}, Y_{t-3}, Y_{t-12},\ \tilde{\mathbf{c}}_{t-1}^\top\right]^\top, \tag{10}$$

where $m_t$ is the calendar month, and $\mathbf{c}_{t-1}$ is a *light* climate vector with at most three lag-1 covariates chosen from rainfall, air temperature, and relative humidity (Section preprocess). This lag-1 restriction is the primary specification to prevent leakage. If an operational pipeline provides reliable same-month climate readings at issue time $t$, a contemporaneous variant is treated as a separate sensitivity analysis under an explicit reporting-delay assumption. Seasonal harmonics and climate are standardized on the *training folds only*; count lags remain unscaled.

Let $\mathcal{I}_h = \{t_0, \ldots, T - h\}$ be the set of training indices after respecting maximal lag and the minimum training length. With $r = \alpha^{-1}$ and $p_{t,h} = r/(r + \mu_{t,h})$, $\mu_{t,h} = \exp(\mathbf{x}_t^\top \beta_h)$, the log-likelihood is

$$\ell(\beta_h, \alpha) = \sum_{t \in \mathcal{I}_h} \left[ \log \Gamma(Y_{t+h} + r) - \log \Gamma(r) - \log \Gamma(Y_{t+h} + 1) + r \log p_{t,h} + Y_{t+h} \log(1 - p_{t,h}) \right]. \tag{11}$$

We estimate $(\beta_h, \alpha)$ by maximum likelihood (standard NB2 GLM fitting; implementation details in Supporting Information). All features in $\mathbf{x}_t$ are computed from information available at issue time $t$; standardization parameters are learned on training folds and applied to validation/test folds.

For a new issue $T$,

$$\widehat{\mu}_{T,h} = \exp(\mathbf{x}_T^\top \widehat{\beta}_h), \qquad \widehat{r} = \widehat{\alpha}^{-1}, \qquad \widehat{p}_{T,h} = \frac{\widehat{r}}{\widehat{r} + \widehat{\mu}_{T,h}}. \tag{14}$$

Point forecasts are $\widehat{\mu}_{T,h}$. Central $(1-\tau) \times 100\%$ prediction intervals use NB quantiles

$$\left[ Q_{NB}\left(\tfrac{\tau}{2}; \widehat{r}, \widehat{p}_{T,h}\right), \ Q_{NB}\left(1 - \tfrac{\tau}{2}; \widehat{r}, \widehat{p}_{T,h}\right) \right], \tag{15}$$

e.g., $\tau = 0.5$ (50%) and $\tau = 0.1$ (90%).

**INGARCH-NB**

The integer-valued GARCH-type model with negative-binomial innovations (INGARCH-NB) adapts volatility-style feedback to count data, capturing short-memory dependence and overdispersion frequently observed in dengue surveillance series [13,22]. We adopt the NB2 mean-variance form to maintain consistency across model families.

Let $\mathcal{F}_{t-1} = \sigma(Y_{t-1}, Y_{t-2}, \ldots)$ be the natural filtration. We assume

$$Y_t \mid \mathcal{F}_{t-1} \sim NB2(\mu_t, \alpha), \qquad \mathrm{Var}(Y_t \mid \mu_t) = \mu_t + \alpha \mu_t^2, \tag{16}$$

with overdispersion $\alpha > 0$. For likelihood evaluation, we map to $(r,p)$ with $r = \alpha^{-1}$ and $p_t = r/(r + \mu_t)$.

We use a log link with seasonal harmonics, observed-count feedback, and conditional-mean feedback:

$$\eta_t \equiv \log \mu_t = b_0 + b_1 \sin\left(\tfrac{2\pi m_t}{12}\right) + b_2 \cos\left(\tfrac{2\pi m_t}{12}\right) + b_3 \log(1 + Y_{t-1}) + b_4 \log\left(1 + \mu_{t-1}\right), \tag{17}$$

so that $\mu_t = \exp(\eta_t)$. The $\log(1 + \cdot)$ transform stabilizes the feedback at zero counts and avoids numerical issues [14]. The term in $\log(1 + \mu_{t-1})$ provides persistence in the conditional mean. Seasonal harmonics capture annual dengue cyclicality. In the main analysis, we omit climate regressors to keep the comparison focused on endogenous dynamics; a climate-augmented variant can be evaluated as a sensitivity check.

Given $y_{1:T}$ and initialization $\mu_0 > 0$, the log-likelihood is

$$\ell(\mathbf{b}, \alpha) = \sum_{t=1}^{T} \left\{ \log \Gamma(Y_t + r) - \log \Gamma(r) - \log \Gamma(Y_t + 1) + r \log p_t + Y_t \log(1 - p_t) \right\}, \tag{18}$$

where $r = \alpha^{-1}$, $p_t = r/(r + \mu_t)$, and $\mu_t$ is generated recursively from (17). We compute the maximum likelihood estimator $\widehat{\theta} = (\widehat{\mathbf{b}}, \widehat{\alpha})$ using box-constrained quasi-Newton (L–BFGS-B), with constraints $\alpha > 0$ and $b_4 \in [0, 1)$ to discourage explosive feedback. In practice, finite-difference gradients with stable initialization (e.g., $\mu_0 = \bar{Y}$) are adequate.

**Multi-horizon forecasting**

To obtain forecasts at horizons $h = 1,2,3$, we use an iterated predictive scheme consistent with the INGARCH recursion. For $h = 1$, the predictive distribution is $Y_{T+1} \mid \mathcal{F}_T \sim NB2(\mu_{T+1}, \alpha)$ with $\mu_{T+1}$ given by (17). For $h > 1$, we propagate uncertainty forward by Monte Carlo simulation: for $b = 1, \ldots, B$, we draw $Y_{T+1}^{(b)}$ from the $h = 1$ predictive distribution, update the recursion to obtain $\mu_{T+2}^{(b)}$, draw $Y_{T+2}^{(b)}$, and continue up to $T+h$. The resulting empirical distribution $\{Y_{T+h}^{(b)}\}_{b=1}^{B}$ defines the probabilistic forecast, from which we compute point forecasts (mean or median), prediction intervals, and proper scoring rules. We use $B$ large enough to stabilize scores (Supporting Information).

## Renewal-NB

We adopt an epidemiological renewal model with an NB2 observation process as a mechanistic baseline [15,23]. At a monthly resolution, the serial-interval kernel should be interpreted as an *effective* kernel that aggregates within-month transmission and reporting delays; we therefore evaluate kernel sensitivity in (Section sensitivity).

Let $\mathcal{F}_{t-1}$ denote the information set up to month $t-1$. Counts follow

$$Y_t \mid \mathcal{F}_{t-1} \sim \text{NB2}(\mu_t, \alpha), \qquad \text{Var}(Y_t \mid \mu_t) = \mu_t + \alpha \mu_t^2, \tag{20}$$

with $r = \alpha^{-1}$ and $p_t = r/(r + \mu_t)$. The discrete renewal equation is

$$\mu_t = R_t \sum_{s=1}^{S} w_s Y_{t-s}, \qquad \sum_{s=1}^{S} w_s = 1, \tag{21}$$

where $w = (w_1, \ldots, w_S)$ is a nonnegative kernel. In the baseline specification, we use $S = 3$ with $w \propto (0.6, 0.3, 0.1)$ as a front-loaded effective kernel at the monthly scale. The effective reproduction number is seasonally modulated,

$$R_t = \exp\left(\gamma_0 + \gamma_1 s_t + \gamma_2 c_t\right), \tag{22}$$

ensuring $R_t > 0$.

**Kernel sensitivity.** To address potential misspecification at a monthly resolution, we evaluate alternative kernel supports and shapes as a sensitivity analysis:

- **Support:** $S \in \{3, 6, 9, 12\}$ months.

- **Shapes:** (i) front-loaded geometric decay $w_s \propto \rho^{s-1}$ with $\rho \in (0, 1)$, and (ii) diffuse kernels (e.g., discretized gamma) normalized to sum to one.

We refit the renewal model under each kernel and compare probabilistic scores and calibration diagnostics to determine whether conclusions about renewal performance are robust to kernel choice.

Given $y_{1:T}$ and support $S$, the renewal recursion is defined for $t > S$. The log-likelihood under NB2 is

$$\ell(\theta) = \sum_{t=S+1}^{T} \left\{ \log \Gamma(Y_t + r) - \log \Gamma(r) - \log \Gamma(Y_t + 1) + r \log p_t + Y_t \log(1 - p_t) \right\}, \tag{24}$$

with $\theta = (\gamma_0, \gamma_1, \gamma_2, \alpha)$, $r = \alpha^{-1}$, $p_t = r/(r + \mu_t)$, and $\mu_t$ given by (21)–(22).

We obtain the MLE $\widehat{\theta} = \arg\max_\theta \ell(\theta)$ via box-constrained L–BFGS–B, with $\alpha > 0$ and a soft bound on seasonal amplitude (e.g., $\gamma_1^2 + \gamma_2^2 \le C$) to prevent unrealistically large forcing. Gradients follow from $\mu_t = R_t \Lambda_t$ where $\Lambda_t = \sum_{s=1}^{S} w_s Y_{t-s}$. In particular,

$$\frac{\partial \mu_t}{\partial \gamma_j} = \mu_t g_j(m_t), \qquad g_0 \equiv 1, \;\; g_1 \equiv \sin\left(\frac{2\pi m_t}{12}\right), \;\; g_2 \equiv \cos\left(\frac{2\pi m_t}{12}\right), \tag{25}$$

and $\partial\ell/\partial\mu_t$ is obtained from (24). In practice, finite-difference derivatives are sufficient due to the low parameter dimension.

**Multi-horizon forecasting.** For $h = 1$, the forecast mean is $\widehat{\mu}_{T+1} = \widehat{R}_{T+1} \sum_{s=1}^{S} w_s Y_{T+1-s}$. For $h > 1$, we generate probabilistic forecasts by iterating the renewal recursion with Monte Carlo simulation: we draw future paths from the NB2

predictive distribution and update the renewal term using simulated counts, yielding an empirical forecast distribution for $Y_{T+h}$. Prediction intervals are computed from the corresponding empirical quantiles.

## BiLSTM-NB

The Bidirectional Long Short-Term Memory model with a Negative-Binomial output head (BiLSTM-NB) couples deep sequence representations with a count likelihood tailored to overdispersed dengue surveillance data. It learns non-linear dependencies while producing horizon-specific predictive distributions suitable for probabilistic evaluation [24,25].

**Inputs and leakage-safe construction.** At a monthly cadence, each training instance at issue time $t$ comprises (i) a univariate count window

$$\mathbf{C}_t \in \mathbb{R}^{W \times 1}, \qquad \mathbf{C}_t = \left[Y_{t-W+1}, \ldots, Y_t\right]^\top, \quad W = 12, \tag{26}$$

and (ii) an auxiliary vector

$$\mathbf{A}_t \in \mathbb{R}^{d_a}, \quad \mathbf{A}_t = \left[s_t, \ c_t, \ Y_{t-1}, Y_{t-2}, Y_{t-3}, Y_{t-12}, \ x_{c,t-1}^{(1)}, \ldots, x_{c,t-1}^{(K_c)}\right]^\top, \tag{27}$$

where $(s_t, c_t)$ seasonal harmonics and autoregressive lags are $\{1,2,3,12\}$, and the "light" climate set uses up to $K_c \le 3$ lag-1 features among rainfall, temperature, and relative humidity (Section climate_timing). To prevent look-ahead, (a) only lagged climate is used, (b) harmonics and climate are standardized on training folds only, and (c) model selection and calibration are performed using data available within each training fold (details below).

**Architecture.** The count window $\mathbf{C}_t$ is passed through two stacked bidirectional LSTM layers with 32 units per direction. Let $\mathbf{h}_{\text{BiLSTM}} \in \mathbb{R}^{64}$ denote the final embedding. A dense block with ReLU activation and dropout (rate 0.2) produces a non-linear summary $\mathbf{h} \in \mathbb{R}^{64}$. To retain a short-memory linear structure, we include an autoregressive skip that maps the four AR lags directly to a horizon-specific mean adjustment. The network concatenates the learned representation with auxiliary features.

$$\mathbf{z}_t = \left[\mathbf{h}^\top, \ \mathbf{A}_t^\top\right]^\top \in \mathbb{R}^{64+d_a}. \tag{28}$$

**Negative-binomial output head.** For each horizon $h \in \{1, 2, 3\}$, the model outputs pre-activations $\eta_{h,t}^{(\mu)}$ and $\eta_{h,t}^{(\alpha)}$ via affine maps of $\mathbf{z}_t$, with an AR-skip term applied to the mean:

$$\eta_{h,t}^{(\mu)} = \mathbf{w}_h^{(\mu)\top}\mathbf{z}_t + b_h^{(\mu)} + s_{h,t}^{\text{AR}}, \qquad \eta_{h,t}^{(\alpha)} = \mathbf{w}_h^{(\alpha)\top}\mathbf{z}_t + b_h^{(\alpha)}, \tag{29}$$

where $s_{h,t}^{\text{AR}} = \theta_h^\top [Y_{t-1}, Y_{t-2}, Y_{t-3}, Y_{t-12}]^\top$. Positivity and numerical stability are enforced with bounded activations:

$$\mu_{h,t} = \text{softplus}(\eta_{h,t}^{(\mu)}), \qquad \alpha_{h,t} = \alpha_{\min} + (\alpha_{\max} - \alpha_{\min})\,\sigma(\eta_{h,t}^{(\alpha)}), \tag{30}$$

with $\alpha_{\min} = 10^{-4}$, $\alpha_{\max} = 2$, and $\sigma(\cdot)$ the logistic sigmoid [26]. We adopt NB2 with mean $\mu_{h,t}$ and variance $\mu_{h,t} + \alpha_{h,t}\mu_{h,t}^2$; equivalently, $r_{h,t} = 1/\alpha_{h,t}$ and $p_{h,t} = r_{h,t}/(r_{h,t} + \mu_{h,t})$.

**Training objective and optimization.** Let $Y_{t+h}$ be the target for horizon $h$. The per-instance multi-horizon negative log-likelihood is

$$\mathcal{L}_t(\Theta) = -\sum_{h=1}^{3} \log \Pr\left(Y = Y_{t+h} \,\middle|\, r_{h,t}, p_{h,t}\right), \tag{31}$$

where $\log \Pr(Y = y \mid r, p) = \log \Gamma(y + r) - \log \Gamma(r) - \log \Gamma(y + 1) + r \log p + y \log(1 - p)$. We optimize with Adam (learning rate $10^{-3}$), gradient-norm clipping ($\|\nabla\| \le 1.0$), early stopping (patience 12 epochs), and `ReduceLROnPlateau` (factor 0.5, patience 6).

**Hyperparameter specification and audit trail.** Given the limited sample size, we pre-specify the BiLSTM-NB configuration *a priori* rather than performing extensive per-fold hyperparameter search. The network uses two BiLSTM layers (32 units per direction), a dense layer (64 units, ReLU) with L2 regularization ($10^{-4}$) and dropout (0.2), and a bounded NB2 dispersion $\alpha \in [10^{-4}, 2]$. Training uses up to 150 epochs, a batch size of 16, early stopping on a time-ordered validation tail, and the learning-rate schedule above. Within each rolling-origin training fold, we reserve the final 20% of the training window (time-ordered) as a validation tail for early stopping and learning-rate scheduling; no test-era observations are used for model selection. All fixed settings, preprocessing rules, and random seeds are reported for auditability. Hyperparameter details are listed in S1 Table.

**Ensembling and calibration.** To stabilize training, we fit an ensemble of $M = 5$ models using fixed seeds and form an equal-weight mixture predictive distribution by averaging the component NB probability mass functions. To improve reliability without look-ahead, calibration is learned using forecasts generated strictly within the training window of each rolling-origin fold (sequentially) and then applied unchanged to the corresponding test issues. Specifically, we apply a monotone isotonic post-processing map to PIT-based CDF values computed from the ensemble mixture, using only training-window forecasts and realized outcomes [26].

## Sensitivity and generalizability analyses

**Renewal kernel sensitivity.** To assess the robustness of the mechanistic baseline at monthly resolution, we refit the Renewal-NB model under alternative kernel supports and shapes: (i) support $S \in \{3, 6, 9, 12\}$ months and (ii) kernel shapes, including front-loaded geometric decay $w_s \propto \rho^{s-1}$ with $\rho \in (0, 1)$ and diffuse kernels (e.g., discretized gamma) normalized to sum to one. We compare probabilistic scores and calibration diagnostics across kernels.

**Climate feature-set sensitivity.** To justify the "light climate" specification, we evaluate an expanded climate feature set (additional lags and/or anomalies) as a sensitivity analysis while preserving leakage-safe timing. Results are reported in the Supporting Information.

**Temporal generalizability.** We assess the stability of conclusions under an era-based evaluation by training on an earlier period and evaluating on a later period (details in Results), using the same leakage-safe rolling-origin protocol within the evaluation era.

## Experimental setup

We evaluate all models under an expanding-window, rolling-origin design for monthly dengue surveillance (January 2015 to December 2024). Let $t$ index months, and let $\mathcal{D}_{1:t}$ denote all data available up to and including the month $t$ (cases, calendar features, and leakage-safe climate covariates). After a minimum training length of 48 months and once all required lagged features are available, each model is refitted $\mathcal{D}_{1:t}$ and issues probabilistic forecasts for horizons $h \in \{1, 2, 3\}$ months ahead, targeting month $t + h$. This procedure repeats for every eligible issue month, yielding a sequence of out-of-sample predictive distributions and realized outcomes for scoring.

For every model and eligible $(t, h)$, we record the predictive NB parameters $(\widehat{\mu}_{t,h}, \widehat{\alpha}_{t,h})$, the implied $(\widehat{r}_{t,h}, \widehat{p}_{t,h})$ with $\widehat{r}_{t,h} = 1/\widehat{\alpha}_{t,h}$ and $\widehat{p}_{t,h} = \widehat{r}_{t,h}/(\widehat{r}_{t,h} + \widehat{\mu}_{t,h})$, the realized outcome $Y_{t+h}$, the log predictive score $\log P(Y_{t+h} \mid \mathcal{D}_{1:t})$, the central 50% and 90% prediction intervals, empirical coverages, interval widths, and randomized PIT values. These per-issue records underpin summary metrics and pairwise Diebold-Mariano tests (Section: metrics).

Because some models may be undefined for certain issue months (e.g., due to lag requirements at the beginning of the series), we report two complementary evaluations: (i) *model-wise* summaries computed on each model's available

issue-target set and (ii) *aligned* comparisons that restrict to the intersection of issue-target pairs shared by all models for a given horizon. The aligned set is used for Diebold-Mariano tests to ensure like-for-like comparisons.

**Evaluation metrics**

We assess probabilistic accuracy, calibration, and sharpness using proper scoring rules and diagnostics for count forecasts [27,28].

**Log score (primary).** For a forecast with NB2 predictive distribution $P(\cdot \mid \widehat{\mu}_i, \widehat{\alpha}_i)$ and observed count $y_i$, the log score is

$$S_i = \log P(Y = y_i \mid \widehat{\mu}_i, \widehat{\alpha}_i). \tag{32}$$

Using the NB2 pmf,

$$P(Y = y \mid \mu, \alpha) = \frac{\Gamma(y + \alpha^{-1})}{\Gamma(\alpha^{-1})\Gamma(y + 1)} \left(\frac{\alpha^{-1}}{\alpha^{-1} + \mu}\right)^{\alpha^{-1}} \left(\frac{\mu}{\alpha^{-1} + \mu}\right)^{y}, \qquad y \in \mathbb{N}_0, \tag{33}$$

the mean log score across $n$ forecasts is

$$\overline{S} = \frac{1}{n}\sum_{i=1}^{n} S_i = \frac{1}{n}\sum_{i=1}^{n} \log P(Y = y_i \mid \widehat{\mu}_i, \widehat{\alpha}_i). \tag{34}$$

Higher $\overline{S}$ indicates better probabilistic accuracy. For presentation as a loss, we also report the negative log score $-S_i$ (smaller is better) [29].

**Predictive interval coverage.** Calibration is assessed using empirical coverage of central prediction intervals at nominal levels $\tau \in \{0.50, 0.90\}$. For each forecast $i$, the equal-tailed interval is

$$\mathrm{PI}_{\tau,i} = \left[F_{\mathrm{NB}}^{-1}\left(\frac{1-\tau}{2}; \widehat{\mu}_i, \widehat{\alpha}_i\right), \, F_{\mathrm{NB}}^{-1}\left(\frac{1+\tau}{2}; \widehat{\mu}_i, \widehat{\alpha}_i\right)\right], \tag{35}$$

where $F_{\mathrm{NB}}^{-1}(\cdot; \mu, \alpha)$ NB denotes the quantile function. The empirical coverage rate is

$$\mathrm{Coverage}_{\tau} = \frac{1}{n}\sum_{i=1}^{n} \mathbb{1}\left[y_i \in \mathrm{PI}_{\tau,i}\right]. \tag{36}$$

Well-calibrated forecasts satisfy $\mathrm{Coverage}_{\tau} \approx \tau$; undercoverage, indicating overconfidence, and overcoverage indicates overly diffuse forecasts [30].

**Median interval width (sharpness).** Sharpness (conditional on calibration) is summarized by the median width of the $\tau$-level interval:

$$\mathrm{MIW}_{\tau} = \mathrm{median}\left\{w_{\tau,i}\right\}_{i=1}^{n}, \tag{37}$$

where

$$w_{\tau,i} = F_{\mathrm{NB}}^{-1}\left(\frac{1+\tau}{2}; \widehat{\mu}_i, \widehat{\alpha}_i\right) - F_{\mathrm{NB}}^{-1}\left(\frac{1-\tau}{2}; \widehat{\mu}_i, \widehat{\alpha}_i\right). \tag{38}$$

Among similarly calibrated models, smaller $\text{MIW}_\tau$ indicates sharper and more informative predictive distributions [31].

**Diebold-Mariano tests with HAC variance.** Pairwise forecast comparisons use the Diebold-Mariano (DM) test for equal predictive accuracy [18]. We define a loss based on the negative log score $L_t = -S_t$. For two competing models with losses $L_{1,t}$ and $L_{2,t}$ non-aligned issue-target pairs, the loss differential is

$$d_t = L_{1,t} - L_{2,t}. \tag{39}$$

The null hypothesis $H_0 : \mathbb{E}[d_t] = 0$ is tested using

$$\text{DM} = \frac{\bar{d}}{\sqrt{\widehat{\sigma}_d^2/n}}, \qquad \bar{d} = \frac{1}{n}\sum_{t=1}^{n} d_t, \tag{40}$$

where $\widehat{\sigma}_d^2$ is a Newey-West heteroskedasticity-and-autocorrelation-consistent (HAC) variance estimator

$$\widehat{\sigma}_d^2 = \widehat{\gamma}_0 + 2\sum_{j=1}^{L}\left(1 - \frac{j}{L+1}\right)\widehat{\gamma}_j, \tag{41}$$

with $\widehat{\gamma}_j = n^{-1}\sum_{t=j+1}^{n}(d_t - \bar{d})(d_{t-j} - \bar{d})$. We set the HAC bandwidth to $L = h - 1$ to reflect serial correlation induced by over-lapping $h$step-ahead forecasts [32]. Two-sided tests use $\alpha = 0.05$; negative DM favors model 1 (lower loss) and positive values favor model 2 [33].

**Randomized PIT histograms.** Calibration is also assessed via probability integral transform (PIT) diagnostics. For discrete predictive distributions, we use randomized PIT values

$$u_i = F_{\text{NB}}(y_i - 1 \mid \widehat{\mu}_i, \widehat{\alpha}_i) + U_i \cdot P(Y = y_i \mid \widehat{\mu}_i, \widehat{\alpha}_i), \tag{42}$$

where $U_i \sim \text{Uniform}(0, 1)$ are independent and $F_{\text{NB}}(y - 1 \mid \mu, \alpha) = P(Y < y \mid \mu, \alpha)$ with $F_{\text{NB}}(-1 \mid \mu, \alpha) = 0$ [34]. Under calibration, $u_i \sim \text{Uniform}(0, 1)$. We visualize the empirical distribution of $\{u_i\}_{i=1}^{n}$ using $B = \lceil\sqrt{n}\rceil$ equal-width bins. Deviations from uniformity indicate: (i) U-shaped histograms (underdispersed forecasts), (ii) inverse-U (overdispersed), (iii) left-skew (systematic overforecasting), and (iv) right-skew (systematic underforecasting). We optionally supplement visual inspection with a uniformity test (e.g., Anderson-Darling), noting that power may be limited for small samples [35].

## Results

We evaluated leakage-safe monthly probabilistic forecasts of dengue cases in Freetown, Sierra Leone, at horizons $h \in \{1, 2, 3\}$. The candidate models comprise NB-GLM, INGARCH-NB, Renewal-NB, a light climate-augmented NB-GLM variant (NB-GLM+Climate), a light climate-informed renewal variant (Renewal+Climate), and a probabilistic BiLSTM with a negative-binomial observation model (BiLSTM-NB). Performance is summarized using mean log score (primary; higher is better), empirical coverage of nominal 50% and 90% prediction intervals (PIs), and median PI widths (sharpness). Unless stated otherwise, headline comparisons use the *global-aligned* issue/target set shared across models within each horizon ($n = 32$ per horizon); additional pairwise-aligned results and model-wise (unaligned) summaries are reported in the Supporting Information.

### Data overview and regime characterization

The cleaned monthly dengue series spans January 2015 to December 2024 (120 months, no missing months). The mean monthly incidence is 20.43 cases, and the variance is 331.73, indicating substantial overdispersion (variance-to-mean

ratio = 16.24; Table data_summary). Zero-count months account for 18.33% observations (22/120), consistent with intermittent transmission at monthly resolution.

To assess performance under heterogeneous transmission intensity, we stratify evaluation targets into *non-outbreak* and *outbreak* regimes using a horizon-specific threshold $thr_h$ applied to the realized target $y_{t+h}$. Specifically, a target month is labeled *outbreak* if $y_{t+h} > thr_h$, and *"non-outbreak"* otherwise. Thresholds are taken from the regime experiment and are similar across horizons ($thr_1 = 33.00$, $thr_2 = 32.50$, $thr_3 = 32.25$). This split isolates high-incidence targets and supports interpretation of calibration and upper-tail behavior during intense transmission periods (Table 1).

## Main aligned probabilistic accuracy across horizons

Table 2 reports performance on the *global-aligned* evaluation set ($n_{aligned} = 32$ per horizon), ensuring a like-for-like comparison across all models. Across horizons, INGARCH-NB attains the best mean log score, indicating the strongest overall distributional accuracy under strict alignment. BiLSTM-NB is consistently competitive and shows strong 90% PI calibration (including perfect coverage at $h = 3$), but it does not exceed INGARCH-NB in mean log score on the global-aligned set.

The NB-GLM baseline undercovers markedly at the 90% level (53.1-62.5% across horizons), consistent with underdispersed predictive distributions under strong overdispersion and changing dynamics. Adding the light climate covariates improves the mean log score relative to NB-GLM at $h = 2$ and $h = 3$, but the resulting forecasts remain substantially under-calibrated (Cover90 $\leq$ 71.9%) and extremely narrow (Width90 $\approx$ 12 cases). Renewal-based baselines exhibit the opposite failure mode: Renewal-NB often attains near-nominal or above-nominal 90% coverage, but typically with wider intervals, which reduces sharpness and penalizes log score at longer horizons. The Renewal+Climate (light) variant becomes particularly diffuse at $h = 2$ (median Width90 = 191), indicating sensitivity of the climate-augmented renewal specification and overly conservative tails in this setting.

Fig 3 visualizes the mean log score ranking by horizon, while Figs 4 and 5 summarize the calibration-sharpness trade-offs consistent with Table 2. Additional diagnostic plots are provided in S1 Fig, S2 Fig and S3 Fig.

## Pairwise significance: Diebold-Mariano tests on aligned samples

To assess whether differences in mean log scores are statistically distinguishable, we use Newey–West/HAC with bandwidth $L = h - 1$, a standard choice for h-step-ahead loss differentials. These samples can be larger than the global-aligned set because alignment is required only between the two models being compared (e.g., $n = 63$ or $n = 69$), rather than across all models simultaneously.

INGARCH-NB significantly outperforms NB-GLM at all horizons ($p = 0.044$ at $h = 1$; $p = 0.036$ at $h = 2$; $p = 0.018$ at $h = 3$), consistent with gains from explicitly modeling conditional-mean dynamics in overdispersed monthly counts. INGARCH-NB also significantly outperforms BiLSTM-NB across horizons ($p \leq 0.034$), indicating that, in this limited-sample monthly setting, the dynamic count model yields a higher average log predictive density than the

**Table 1. Dataset summary for monthly dengue cases in Freetown, Sierra Leone (cleaned series).**

| Date range | 2015–01–2024–12 |
|---|---|
| Total months | 120 |
| Observed months | 120 |
| Missing months | 0 |
| Zero-count months (%) | 22 (18.33%) |
| Mean cases | 20.43 |
| Variance | 331.73 |
| Overdispersion (Var/Mean) | 16.24 |

PLOS Global Public Health

**Table 2. Global-aligned probabilistic performance by horizon (higher mean log score is better).**

| Model | Horizon | $n_{aligned}$ | Mean log score | Cover50 (%) | Cover90 (%) | Width90 |
|---|---|---|---|---|---|---|
| INGARCH–NB | 1 | 32 | -3.716 | 59.4 | 87.5 | 53.0 |
| BiLSTM–NB | 1 | 32 | -3.771 | 65.6 | 90.6 | 52.0 |
| Renewal–NB | 1 | 32 | -3.882 | 71.9 | 96.9 | 66.5 |
| NB–GLM+Climate (light) | 1 | 32 | -3.882 | 25.0 | 62.5 | 14.5 |
| NB–GLM | 1 | 32 | -4.118 | 28.1 | 56.2 | 16.5 |
| Renewal+Climate (light) | 1 | 32 | -4.196 | 75.0 | 87.5 | 74.0 |
| INGARCH–NB | 2 | 32 | -3.582 | 53.1 | 96.9 | 32.0 |
| NB–GLM+Climate (light) | 2 | 32 | -3.694 | 25.0 | 71.9 | 12.0 |
| NB–GLM | 2 | 32 | -3.798 | 31.2 | 62.5 | 12.0 |
| BiLSTM–NB | 2 | 32 | -3.979 | 65.6 | 96.9 | 57.0 |
| Renewal–NB | 2 | 32 | -4.107 | 40.6 | 96.9 | 39.5 |
| Renewal+Climate (light) | 2 | 32 | -4.989 | 53.1 | 78.1 | 191.0 |
| INGARCH–NB | 3 | 32 | -3.604 | 53.1 | 87.5 | 30.5 |
| BiLSTM–NB | 3 | 32 | -3.660 | 53.1 | 100.0 | 69.5 |
| NB–GLM+Climate (light) | 3 | 32 | -4.038 | 31.2 | 62.5 | 12.5 |
| Renewal–NB | 3 | 32 | -4.152 | 43.8 | 96.9 | 124.5 |
| NB–GLM | 3 | 32 | -4.162 | 28.1 | 53.1 | 13.5 |
| Renewal+Climate (light) | 3 | 32 | -4.848 | 28.1 | 78.1 | 81.5 |

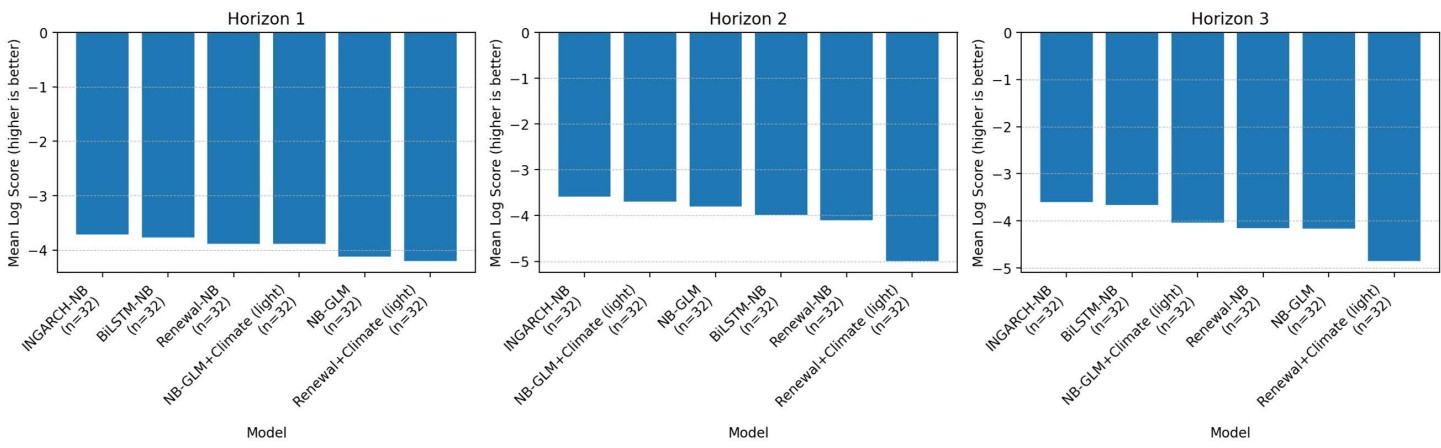

**Fig 3. Global-aligned mean log score by model and horizon (higher is better).** Barplots of the mean log score on the global-aligned set ($n = 32$ per horizon). INGARCH-NB ranks best across horizons, with BiLSTM-NB competitive; renewal-based variants are penalized by diffuse distributions, and GLM variants by under-dispersion.

deep sequence model. Conversely, BiLSTM-NB substantially improves log score relative to Renewal+Climate (light) at all horizons ($p \leq 0.020$), consistent with the renewal climate variant producing overly diffuse forecasts (notably at longer horizons) that are penalized under log scoring. Fig 6 summarizes the count of statistically significant wins by horizon (Table 3).

Model Performance — 90% Coverage (%) (closer to 90 is better)

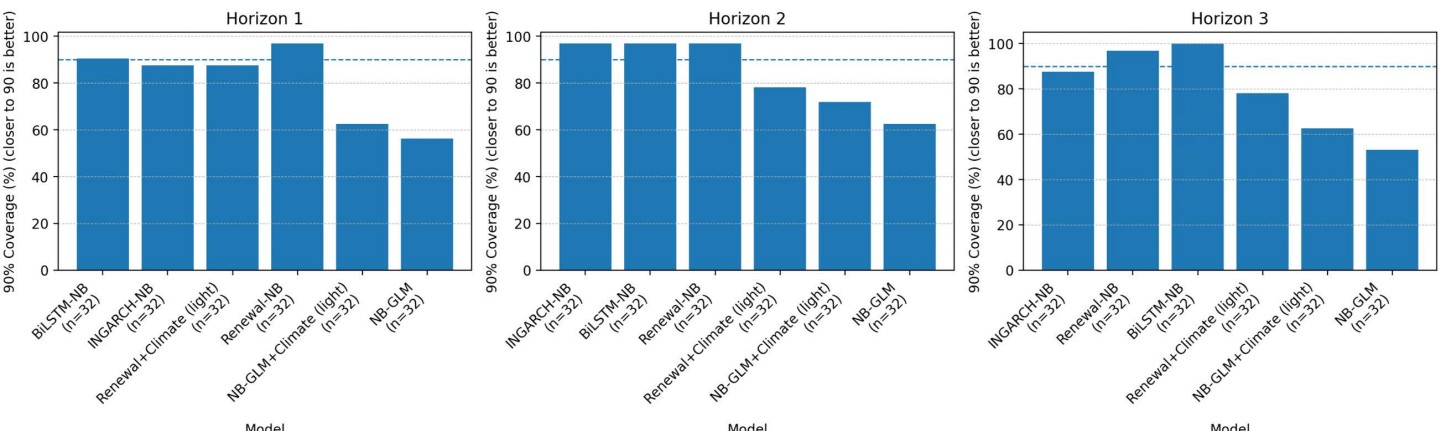

**Fig 4. Global-aligned 90% PI coverage by model and horizon (closer to 90% is better).** Coverage of nominal 90% predictive intervals on the global-aligned set. GLM variants under-cover substantially; BiLSTM-NB attains very high coverage at longer horizons; INGARCH-NB maintains generally good calibration without extreme width inflation.

Model Performance — Median 90% Interval Width (smaller is better)

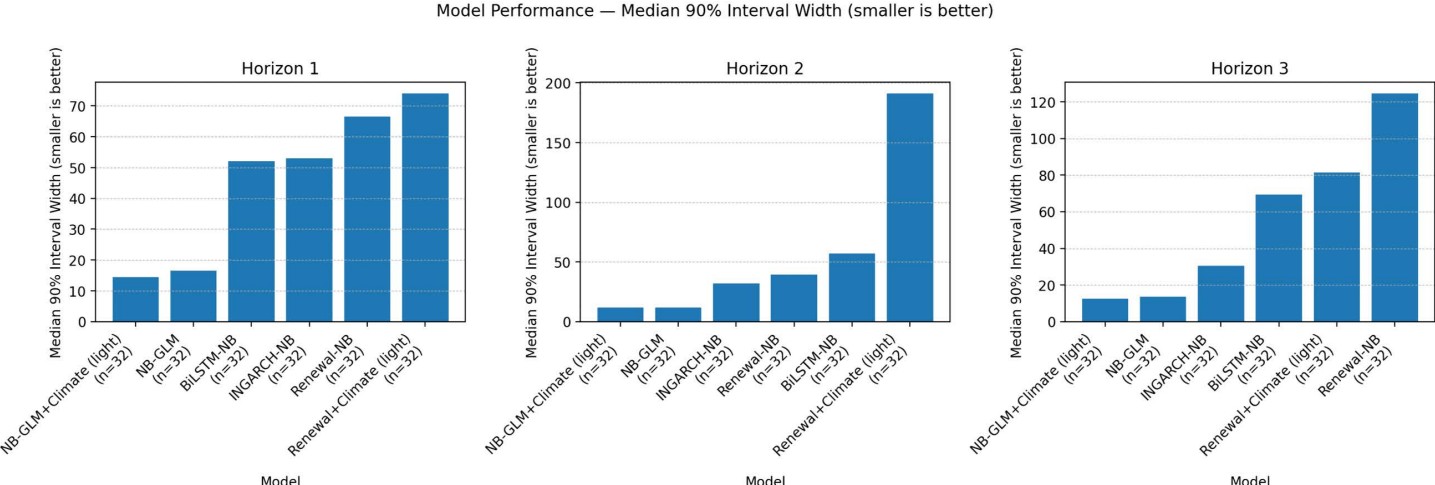

**Fig 5. Global-aligned median 90% PI width by model and horizon (smaller is sharper).** Median 90% PI widths highlight uncertainty inflation for renewal-based variants at longer horizons. GLM variants remain narrow but undercover; INGARCH-NB offers a better calibration-sharpness trade-off.

## Calibration and sharpness of predictive intervals

Operational utility depends on joint calibration (coverage) and sharpness (interval width). The NB-GLM baselines are the sharpest, producing the narrowest 90% predictive intervals across horizons (median width $\approx$ 12–17 cases), but they markedly under-cover at the 90% level (about 53–62% for NB-GLM and 62–72% for NB-GLM+Climate on the global-aligned set), indicating systematic overconfidence. INGARCH-NB provides the best overall log-score accuracy while maintaining generally good calibration without extreme width inflation; this is most evident at $h=2$, where 90% coverage reaches 96.9% with a relatively compact median 90% width of 32 cases.

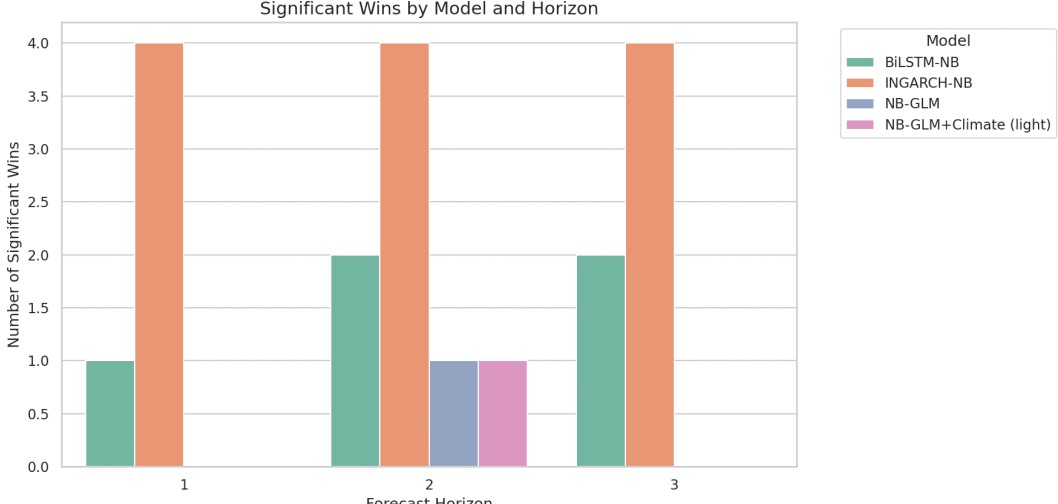

**Fig 6. Significant wins by model and horizon (DM tests, $p < 0.05$).** Count of pairwise comparisons in which each model significantly improves mean log score over another model under HAC-robust DM testing. INGARCH-NB achieves the most consistent significant wins across horizons.

**Table 3. Focused Diebold-Mariano tests on pairwise-aligned samples (log score differences). Positive mean difference favors Model 1.**

| Model 1 | Model 2 | Horizon | $n$ | Mean diff | $p$-value |
|---|---|---|---|---|---|
| INGARCH–NB | NB–GLM | 1 | 69 | 0.425 | 0.044 |
| INGARCH–NB | NB–GLM | 2 | 69 | 0.517 | 0.036 |
| INGARCH–NB | NB–GLM | 3 | 69 | 0.537 | 0.018 |
| INGARCH–NB | BiLSTM–NB | 1 | 63 | 0.197 | 0.027 |
| INGARCH–NB | BiLSTM–NB | 2 | 63 | 0.258 | 0.004 |
| INGARCH–NB | BiLSTM–NB | 3 | 63 | 0.188 | 0.034 |
| BiLSTM–NB | Renewal+Climate (light) | 1 | 32 | 0.425 | 0.020 |
| BiLSTM–NB | Renewal+Climate (light) | 2 | 32 | 1.010 | 0.001 |
| BiLSTM–NB | Renewal+Climate (light) | 3 | 32 | 1.189 | 0.000 |

BiLSTM-NB attains a high 90% coverage, reaching 100% at $h = 3$, but does so with wider intervals (median 90% width 69.5), consistent with more conservative upper-tail behavior. Renewal-based models illustrate a common failure mode in discrete probabilistic forecasting: achieving nominal (or near-nominal) coverage by inflating uncertainty. Renewal-NB produces substantially wider intervals at longer horizons (e.g., median 90% width 124.5 at $h = 3$), and Renewal+Climate (light) becomes highly diffuse at $h = 2$ (median 90% width 191.0), which is strongly penalized under log scoring and limits practical decision usefulness despite occasional adequate coverage. Fig 7 summarizes the accuracy-calibration-sharpness trade-off across horizons.

## Seasonality of skill: month-of-year log score patterns

To probe whether relative forecast skill varies across the calendar year, we computed month-of-year mean log scores on the aligned sample for INGARCH-NB and BiLSTM-NB (Figs 8–10). This analysis is descriptive because the per-month sample sizes are small (typically $\approx$ 5–6 aligned forecasts per month), so month-to-month fluctuations should not be over-interpreted. Nevertheless, a consistent seasonal structure is visible across horizons: both models achieve their best

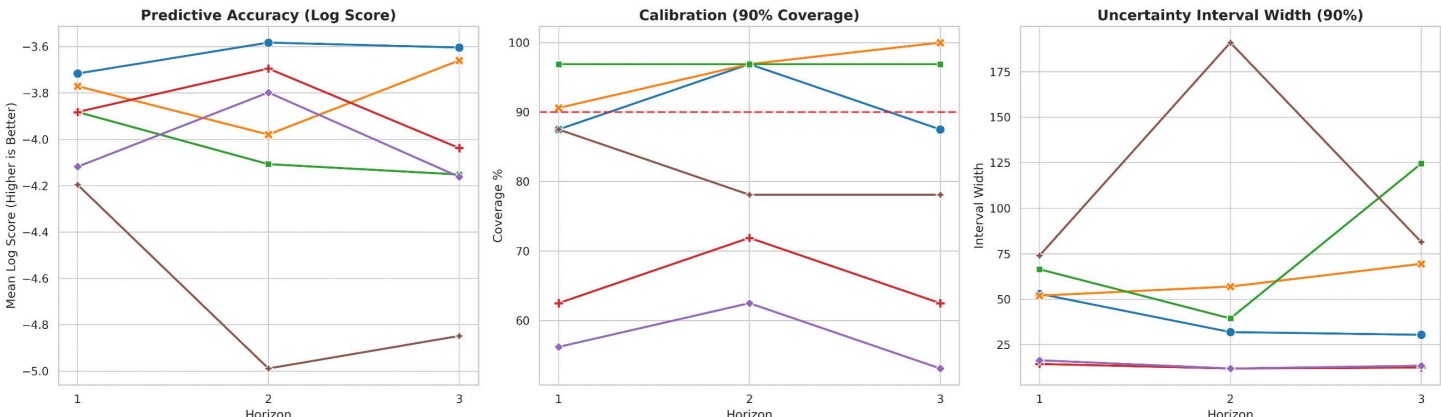

**Fig 7. Global-aligned overview: accuracy, calibration, and width across horizons.** Line summaries across horizons showing the accuracy-calibration-sharpness trade-off. INGARCH-NB is consistently strong in log score with generally good calibration and moderate widths; GLM variants under-cover; renewal variants inflate width at longer horizons.

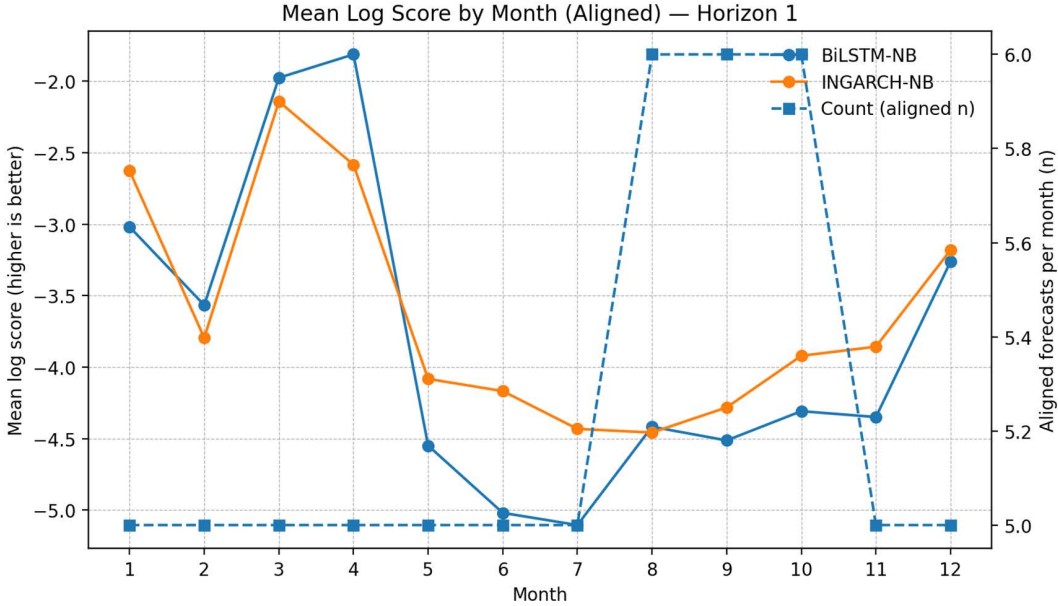

**Fig 8. Seasonal log score patterns (aligned): BiLSTM-NB vs INGARCH-NB at $h=1$.** Month-of-year mean log scores on the aligned set (small per-month counts shown). Both models perform best around months 3–4 and deteriorate mid-year; INGARCH-NB is typically more stable across months.

(least negative) log scores in late Q1/early Q2 (roughly months 3–4), while performance tends to deteriorate around mid-year (approximately months 6–8). This mid-year degradation is consistent with periods in which transmission intensity and dispersion may shift more abruptly, making distributional forecasting more challenging at monthly aggregations.

Across months, INGARCH-NB appears more stable, with smaller swings in mean log score and fewer sharp deteriorations in mid-to-late year, whereas BiLSTM-NB shows occasional month-specific advantages (notably around months 3–4) but with greater variability. Given the limited counts per month, these patterns are best viewed as qualitative diagnostics that complement the globally aligned summaries rather than as definitive evidence of month-specific dominance.

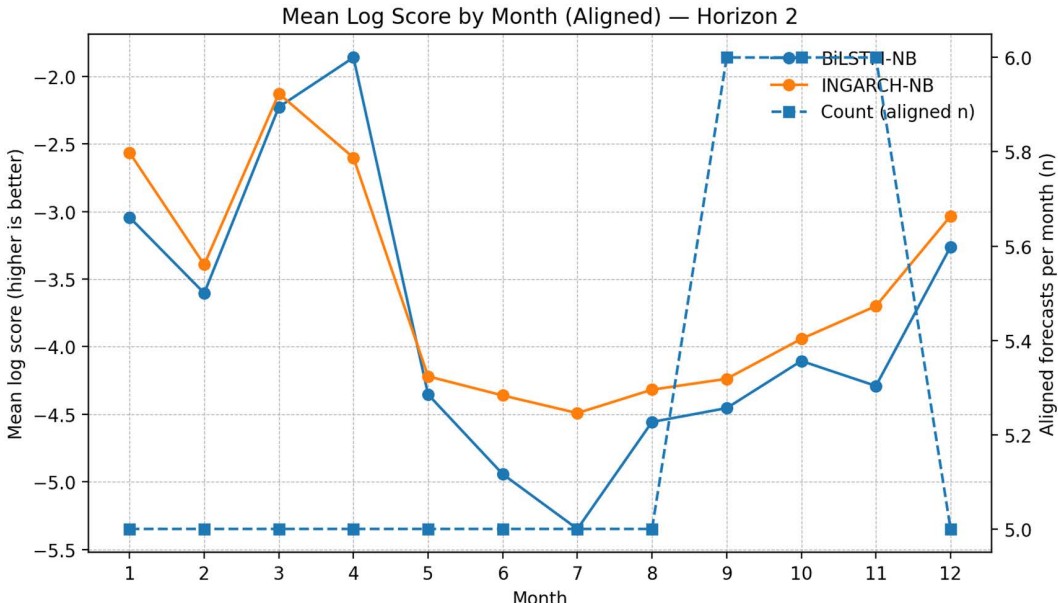

**Fig 9. Seasonal log score patterns (aligned): BiLSTM-NB vs INGARCH-NB at _h_ = 2.** Month-of-year comparison at _h_ = 2 shows similar seasonal structure and mid-year difficulty; interpret descriptively due to small per-month sample sizes.

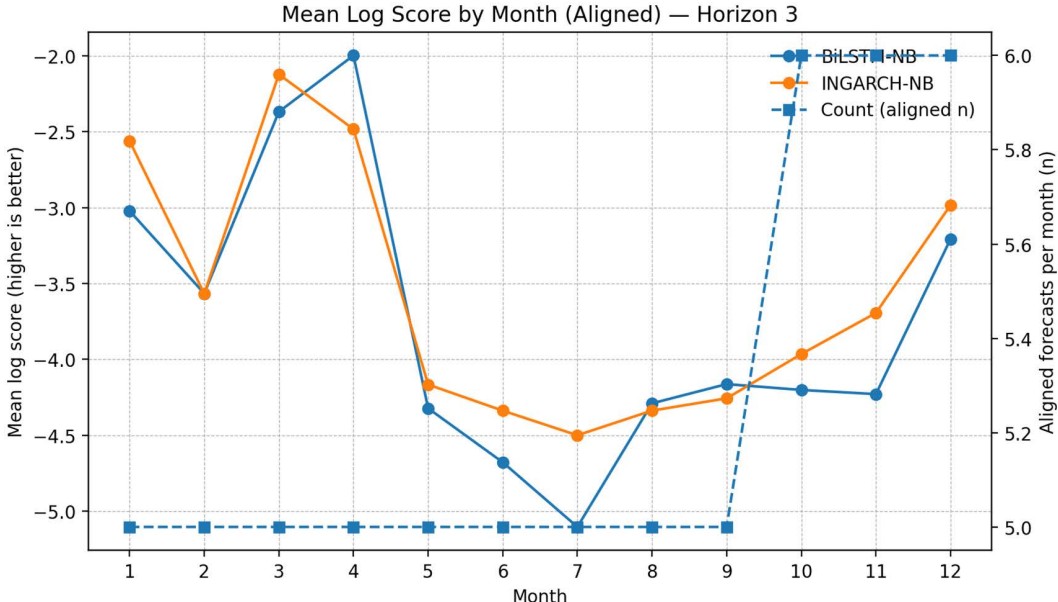

**Fig 10. Seasonal log score patterns (aligned): BiLSTM-NB vs INGARCH-NB at _h_ = 3** month-of-year comparison at _h_ = 3 shows persistent mid-year performance degradation; INGARCH-NB generally remains less variable than BiLSTM-NB.

## Climate signal contribution: the "light climate" experiment

To justify the minimal climate feature set, we evaluated a leakage-safe *light climate* design using only covariates available at the issue time under conservative timing: lag-1 precipitation, temperature, and relative humidity. Table 4 summarizes a within-family climate ablation for NB-GLM and renewal models on aligned evaluation months.

For NB-GLM, adding light climate features yields small improvements in mean log score, with only borderline evidence at $h = 3$ (DM $p = 0.0978$), and no statistically significant gains at the 5% level across horizons (DM $p = 0.312, 0.312, 0.098$ for $h = 1,2,3$). This indicates that while lagged climate may carry some predictive signal, the restricted lag-1-only specification and the linear predictor are insufficient to translate it into reliably improved probabilistic accuracy under strict leakage control. For the mechanistic renewal model, the light climate fit does not improve performance: DM tests favor the non-climate renewal specification at all horizons ($p > 0.35$), and Renewal + Climate forecasts are typically less sharp, suggesting that the additional climate forcing can destabilize tails (width inflation) without delivering commensurate gains in log-score accuracy. At a monthly cadence, the renewal specification is sensitive to kernel assumptions; mis-specification can manifest as tail inflation

## Outbreak vs non-outbreak performance: regime-stratified analysis

To assess robustness under heterogeneous transmission intensity, we stratified the aligned evaluation targets into *non-outbreak* and *outbreak* regimes using the horizon-specific thresholds $\text{thr}_h$ (Table 5). Because the outbreak subset is small (typically $n = 6 - 8$ per horizon), regime-specific rankings should be interpreted as *descriptive* rather than definitive; a few extreme months can materially shift averages.

## Non-outbreak regime

In non-outbreak months, INGARCH-NB yields the best mean log score across horizons ($h = 1$: −3.52; $h = 2$: −3.31; $h = 3$: −3.32), with BiLSTM-NB close behind (Table 5). This pattern is consistent with conditional mean dynamics capturing most of the predictive signal when incidence is moderate. Calibration is acceptable for INGARCH-NB in non-outbreak months (90% coverage = 84.6% − 95.8%), while BiLSTM-NB is more conservative (often reaching 100% 90% coverage) but at wider uncertainty. GLM-based models remain sharp but under the cover in non-outbreak months, indicating persistent overconfidence even outside outbreaks.

## Outbreak regime: accuracy-sharpness trade-offs dominate

During outbreaks, rankings differ, and interpretation hinges on the accuracy-sharpness trade-off. GLM-based models can achieve competitive mean log scores at $h = 1 - 2$ with comparatively moderate widths (e.g., NB-GLM+Climate at $h = 1$: mean log score −3.59, 90% coverage = 83.3%, width90 = 23.0; Table 5). In contrast, INGARCH-NB and BiLSTM-NB often avoid tail misses in outbreaks primarily by issuing much wider intervals (e.g., at $h = 1$, width90 = 126.5 for INGARCH-NB

**Table 4. Climate ablation (light climate; aligned on evaluation months). Light climate uses only lag-1 precipitation, temperature, and humidity to remain leakage-safe and deployable when real-time climate products are limited.**

| Comparison | *h* | *n* | Better | *t* | *p* | Comment |
|---|---|---|---|---|---|---|
| GLM+Climate vs GLM | 1 | 35 | GLM+Climate | 1.011 | 0.3121 | small gain, not significant |
| GLM+Climate vs GLM | 2 | 35 | GLM+Climate | 1.011 | 0.3122 | small gain, not significant |
| GLM+Climate vs GLM | 3 | 35 | GLM+Climate | 1.656 | 0.0978 | borderline evidence at $h=3$ |
| Renewal+Climate vs Renewal | 1 | 33 | Renewal | -0.679 | 0.4974 | climate fit not beneficial |
| Renewal+Climate vs Renewal | 2 | 33 | Renewal | -0.920 | 0.3576 | climate fit not beneficial |
| Renewal+Climate vs Renewal | 3 | 33 | Renewal | -0.872 | 0.3831 | climate fit not beneficial |

**Table 5. Regime-stratified probabilistic performance by horizon (aligned evaluation). Outbreak months are defined by $y_{t+h} > thr_h$, where $thr_h$ is the horizon-specific threshold used in the regime split. Tail-miss rate is the percentage of times $y_{t+h}$ exceeds the upper 90% predictive interval bound.**

| h | Regime | Model | n | Mean log | Cover 90% | Width 90% | Tail miss |
|---|--------|-------|---|----------|-----------|-----------|-----------|
| **Panel A: Horizon h=1 (thr$_1$=33.00)** | | | | | | | |
| 1 | Non-outbreak | INGARCH-NB | 26 | -3.521816 | 84.6 | 14.5 | 11.5 |
| 1 | Non-outbreak | BiLSTM-NB | 26 | -3.581115 | 88.5 | 48.0 | 11.5 |
| 1 | Non-outbreak | Renewal-NB | 26 | -3.588563 | 96.2 | 40.5 | 3.8 |
| 1 | Non-outbreak | Renewal+Climate (light) | 26 | -3.864421 | 84.6 | 37.5 | 15.4 |
| 1 | Non-outbreak | NB-GLM+Climate (light) | 26 | -3.949914 | 57.7 | 10.5 | 11.5 |
| 1 | Non-outbreak | NB-GLM | 26 | -4.053269 | 57.7 | 8.0 | 11.5 |
| 1 | Outbreak | NB-GLM+Climate (light) | 6 | -3.588116 | 83.3 | 23.0 | 0.0 |
| 1 | Outbreak | NB-GLM | 6 | -4.398377 | 50.0 | 23.5 | 16.7 |
| 1 | Outbreak | INGARCH-NB | 6 | -4.557375 | 100.0 | 126.5 | 0.0 |
| 1 | Outbreak | BiLSTM-NB | 6 | -4.591249 | 100.0 | 89.5 | 0.0 |
| 1 | Outbreak | Renewal-NB | 6 | -5.153187 | 100.0 | 363.0 | 0.0 |
| 1 | Outbreak | Renewal+Climate (light) | 6 | -5.632359 | 100.0 | 1061.0 | 0.0 |
| **Panel B: Horizon h=2 (thr$_2$=32.50)** | | | | | | | |
| 2 | Non-outbreak | INGARCH-NB | 24 | -3.309229 | 95.8 | 22.0 | 4.2 |
| 2 | Non-outbreak | Renewal-NB | 24 | -3.637637 | 95.8 | 30.5 | 4.2 |
| 2 | Non-outbreak | BiLSTM-NB | 24 | -3.674250 | 100.0 | 54.5 | 0.0 |
| 2 | Non-outbreak | NB-GLM+Climate (light) | 24 | -3.719250 | 70.8 | 10.5 | 12.5 |
| 2 | Non-outbreak | NB-GLM | 24 | -3.811939 | 58.3 | 11.0 | 25.0 |
| 2 | Non-outbreak | Renewal+Climate (light) | 24 | -4.742912 | 70.8 | 11.5 | 29.2 |
| 2 | Outbreak | NB-GLM+Climate (light) | 8 | -3.619905 | 75.0 | 22.5 | 0.0 |
| 2 | Outbreak | NB-GLM | 8 | -3.755299 | 75.0 | 23.0 | 0.0 |
| 2 | Outbreak | INGARCH-NB | 8 | -4.401874 | 100.0 | 86.0 | 0.0 |
| 2 | Outbreak | BiLSTM-NB | 8 | -4.893744 | 87.5 | 68.0 | 12.5 |
| 2 | Outbreak | Renewal-NB | 8 | -5.515626 | 100.0 | 564.0 | 0.0 |
| 2 | Outbreak | Renewal+Climate (light) | 8 | -5.726214 | 100.0 | 731.0 | 0.0 |
| **Panel C: Horizon h=3 (thr$_3$=32.25)** | | | | | | | |
| 3 | Non-outbreak | INGARCH-NB | 24 | -3.317427 | 87.5 | 16.5 | 8.3 |
| 3 | Non-outbreak | BiLSTM-NB | 24 | -3.384828 | 100.0 | 71.0 | 0.0 |
| 3 | Non-outbreak | Renewal-NB | 24 | -3.587956 | 95.8 | 25.5 | 4.2 |
| 3 | Non-outbreak | NB-GLM+Climate (light) | 24 | -3.950630 | 62.5 | 8.5 | 12.5 |
| 3 | Non-outbreak | NB-GLM | 24 | -4.179056 | 50.0 | 8.0 | 12.5 |
| 3 | Non-outbreak | Renewal+Climate (light) | 24 | -4.211067 | 79.2 | 39.0 | 20.8 |
| 3 | Outbreak | NB-GLM | 8 | -4.109809 | 62.5 | 21.5 | 12.5 |
| 3 | Outbreak | NB-GLM+Climate (light) | 8 | -4.299347 | 62.5 | 21.5 | 12.5 |
| 3 | Outbreak | INGARCH-NB | 8 | -4.462550 | 87.5 | 102.0 | 12.5 |
| 3 | Outbreak | BiLSTM-NB | 8 | -4.483715 | 100.0 | 62.5 | 0.0 |
| 3 | Outbreak | Renewal-NB | 8 | -5.842629 | 100.0 | 1032.0 | 0.0 |
| 3 | Outbreak | Renewal+Climate (light) | 8 | -6.759668 | 75.0 | 2563.5 | 25.0 |

and =89.5 for BiLSTM-NB), which reduces sharpness and can lower log score unless the realized count falls deep in the upper tail. Therefore, outbreak-month comparisons should not be judged on coverage alone: very high coverage can reflect interval inflation rather than well-targeted uncertainty.

### Renewal models show instability under outbreaks

Renewal-NB and especially Renewal+Climate (light) exhibit the most extreme outbreak behavior, producing very large interval widths (e.g., Renewal-NB width90 = 363.0 at $h=1$, = 564.0 at $h=2$; Renewal+Climate width90 = 1061.0 at $h=1$, = 731.0 at $h=2$, and = 2563.5 at $h=3$) alongside poor mean log scores (Table 5). The near-zero tail-miss rates in several outbreak cells are therefore not evidence of superior calibration; they largely reflect over-diffuse predictive distributions that sacrifice sharpness.

### Visual summary of regime effects

Fig 11 summarizes the mean log score by horizon and regime and highlights how some models degrade disproportionately in outbreaks. Complementary horizon-wise regime dashboards (S4-S6 Figs) break down (i) mean log score, (ii) 50%/90% coverage, (iii) interval widths, and (iv) upper-tail miss rates, reinforcing that outbreak performance must be assessed jointly on *calibration and sharpness* rather than coverage alone. Additional diagnostic plots are provided in S3 Fig, S4 Fig, and S5 Fig.

### Generalizability across time: era-based evaluation (2021–2024)

To probe temporal robustness under potential distribution shift, we performed an era-based evaluation by restricting scoring to targets in 2021–2024 and recomputing *aligned* probabilistic metrics across all models and horizons. Because the light-climate models are available only on a sparser set of issue dates, the aligned intersection for this era is smaller than in the full-period analysis, yielding $n_{aligned} = 23$ at $h=1$ and $n_{aligned} = 24$ at $h=2,3$ common forecast cases across all models. Table 6 summarizes log-score accuracy (higher is better; values are negative because they are log probabilities) together with interval calibration and sharpness diagnostics.

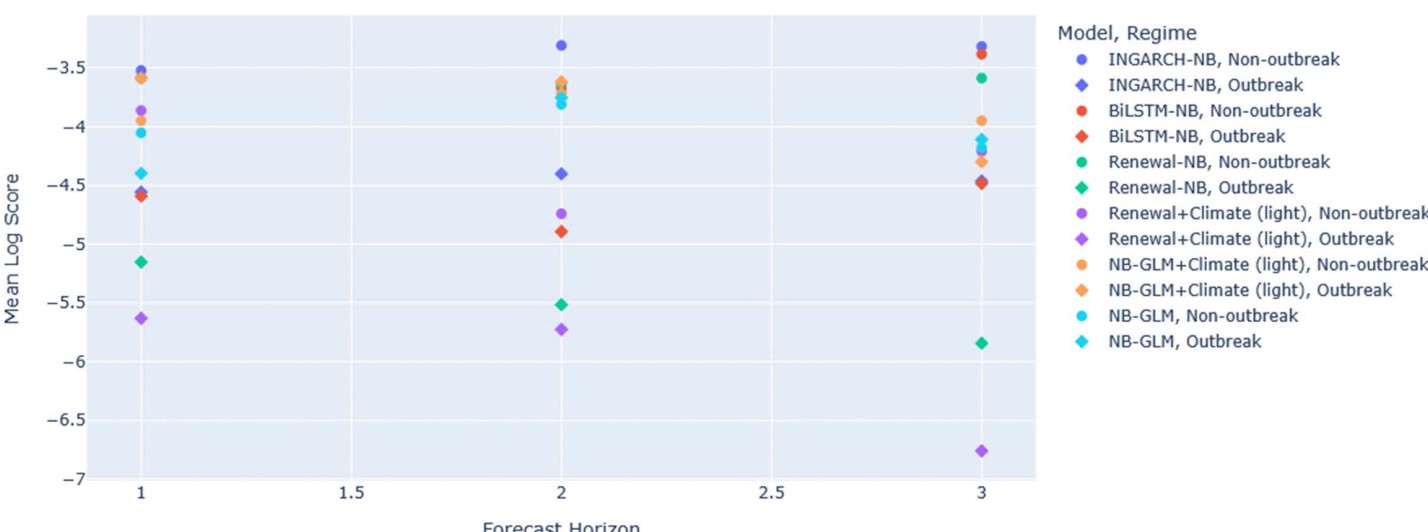

**Fig 11. Regime-stratified mean log score vs horizon.** Points show mean log score by model, horizon, and regime (non-outbreak vs outbreak). Higher (less negative) is better. The outbreak subset is small, so dispersion across models is expected.

**Table 6. Era-based aligned performance on targets in 2021–2024. Aligned intersection across all models in this era yields $n_{aligned}=23$ at $h=1$ and $n_{aligned}=24$ at $h=2,3$. A higher mean log score indicates better probabilistic accuracy. Coverage is empirical PI coverage; widths are median PI widths.**

| Model | h | n | Mean log score | Cover 50% | Cover 90% | Width 50% | Width 90% |
|---|---|---|---|---|---|---|---|
| BiLSTM-NB | 1 | 23 | -3.5856 | 60.9 | 91.3 | 20.0 | 49.0 |
| INGARCH-NB | 1 | 23 | -3.7147 | 56.5 | 87.0 | 21.0 | 52.0 |
| Renewal-NB | 1 | 23 | -3.8236 | 65.2 | 100.0 | 25.0 | 69.0 |
| Renewal+Climate (light) | 1 | 23 | -4.1783 | 73.9 | 82.6 | 14.0 | 43.0 |
| NB-GLM+Climate (light) | 1 | 23 | -4.1793 | 17.4 | 56.5 | 6.0 | 15.0 |
| NB-GLM | 1 | 23 | -4.1850 | 26.1 | 52.2 | 7.0 | 16.0 |
| INGARCH-NB | 2 | 24 | -3.5504 | 50.0 | 100.0 | 11.5 | 29.0 |
| NB-GLM+Climate (light) | 2 | 24 | -3.7487 | 25.0 | 70.8 | 5.0 | 12.0 |
| NB-GLM | 2 | 24 | -3.8149 | 29.2 | 62.5 | 4.5 | 12.0 |
| BiLSTM-NB | 2 | 24 | -3.9630 | 62.5 | 100.0 | 21.5 | 54.5 |
| Renewal-NB | 2 | 24 | -4.1188 | 54.2 | 95.8 | 23.5 | 63.5 |
| Renewal+Climate (light) | 2 | 24 | -4.8276 | 50.0 | 75.0 | 32.5 | 101.0 |
| BiLSTM-NB | 3 | 24 | -3.5592 | 45.8 | 100.0 | 23.0 | 68.5 |
| INGARCH-NB | 3 | 24 | -3.6423 | 50.0 | 83.3 | 12.5 | 30.5 |
| Renewal-NB | 3 | 24 | -4.1274 | 41.7 | 100.0 | 47.5 | 130.0 |
| NB-GLM | 3 | 24 | -4.2386 | 25.0 | 50.0 | 5.0 | 12.5 |
| NB-GLM+Climate (light) | 3 | 24 | -4.2852 | 29.2 | 62.5 | 5.5 | 12.0 |
| Renewal+Climate (light) | 3 | 24 | -4.8068 | 25.0 | 75.0 | 22.5 | 69.0 |

Across 2021–2024, the main conclusions persist: INGARCH-NB and BiLSTM-NB remain the strongest performers, while GLM-based models yield the narrowest intervals but tend to under-cover, indicating overconfidence. However, restricting to this era induces some horizon-specific reordering among the top methods. BiLSTM-NB attains the best mean log score at $h=1$ (mean −3.586) and $h=3$ (mean −3.559), whereas INGARCH-NB is best at $h=2$ (mean −3.550) and remains comparatively sharp at longer horizons (e.g., width90 = 30.5 at $h=3$). Renewal-NB remains well calibrated in this era (90% coverage equals 100% at $h=1$ and $h=3$, and 95.8% at $h=2$) but is less sharp, with wider median predictive intervals, especially at $h=3$ (width90 = 130.0).

Pairwise Diebold-Mariano (DM) tests on the same 2021–2024 aligned subset (Table 7) indicate that some differences remain statistically distinguishable despite the reduced sample size. At $h=2$, INGARCH-NB outperforms BiLSTM-NB with strong evidence (DM $p < 10^{-4}$). At $h=1$, BiLSTM-NB outperforms Renewal+Climate (light) (DM $p=0.0106$). At $h=3$, BiLSTM-NB outperforms Renewal-NB (DM $p=0.0037$), and INGARCH-NB slightly outperforms Renewal-NB (DM $p=0.0481$). Other pairwise differences are directionally consistent but not statistically significant at conventional thresholds, which is expected given the smaller aligned sample induced by the climate-available subset.

**Table 7. Focused Diebold-Mariano tests on 2021-2024 aligned subset. Positive DM $t$ indicates Model 1 has a higher log score (better).**

| Model 1 | Model 2 | h | n | DM $t$ | p |
|---|---|---|---|---|---|
| INGARCH-NB | BiLSTM-NB | 2 | 24 | 5.481 | < $10^{-4}$ |
| Renewal-NB | BiLSTM-NB | 3 | 24 | -2.903 | 0.0037 |
| Renewal+Climate (light) | BiLSTM-NB | 1 | 23 | -2.555 | 0.0106 |
| INGARCH-NB | Renewal–NB | 3 | 24 | 1.976 | 0.0481 |

Fig 12 visualizes the mean log score ranking by horizon in 2021–2024, while Fig 13 summarizes calibration and sharpness on the same aligned subset.

## Discussion

In resource-constrained settings, forecasts of infectious disease burden increasingly guide preparedness, inform vector-control timing, and support risk communication. Using leakage-safe monthly dengue surveillance from Freetown, Sierra Leone (2015–2024), we evaluated a spectrum of probabilistic forecasting approaches—including a statistical regression baseline (NB-GLM), a dynamic count model (INGARCH–NB), mechanistic renewal models (Renewal-NB and a light climate-informed variant), and a deep sequence model with a negative binomial output (BiLSTM-NB)—under a harmonized expanding-window, rolling-origin design. Our analysis yielded three findings with direct relevance to applied public health. First, under strict global alignment across all models, INGARCH-NB delivered the strongest overall distributional accuracy (highest mean log score) across horizons $h \in \{1, 2, 3\}$, suggesting that parsimonious conditional mean dynamics can be highly effective at a monthly cadence. Second, BiLSTM-NB was consistently competitive and exhibited strong reliability at the 90% level (notably reaching 100% coverage at $h = 3$ in the aligned set) but achieved this with wider uncertainty, reflecting more conservative tail behavior. Third, renewal-based specifications were sensitive in this setting: while Renewal-NB often achieved high nominal coverage, it did so with markedly inflated interval widths at longer horizons, and the light climate-informed renewal variant could become extremely diffuse, leading to poor log scores and limited operational sharpness.

**Implications for operational decision-making.** Public health programs balance *near-term responsiveness* (clinical readiness, targeted mobilization) with *medium-range planning* (vector-control campaigns, community engagement ahead of seasonal upswings). Our results suggest that selecting a single "best" model is inadvisable without considering the

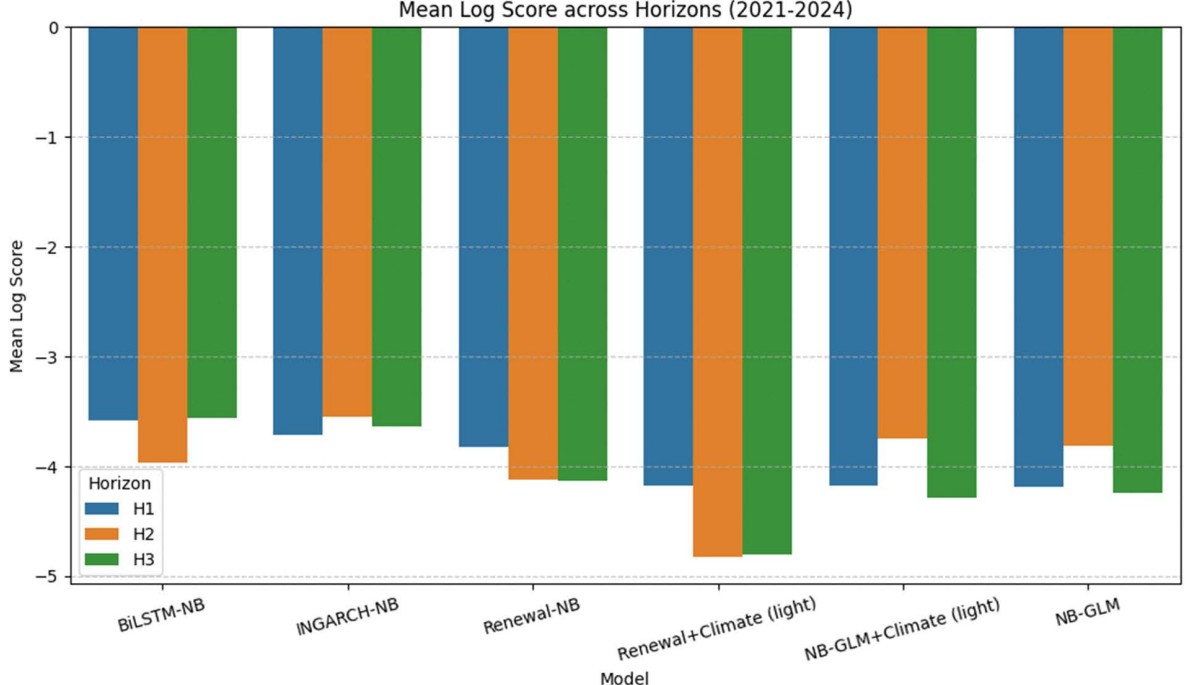

**Fig 12. Era-based predictive accuracy (2021-2024).** Mean log score by model and horizon on the 2021-2024 aligned subset (higher is better).

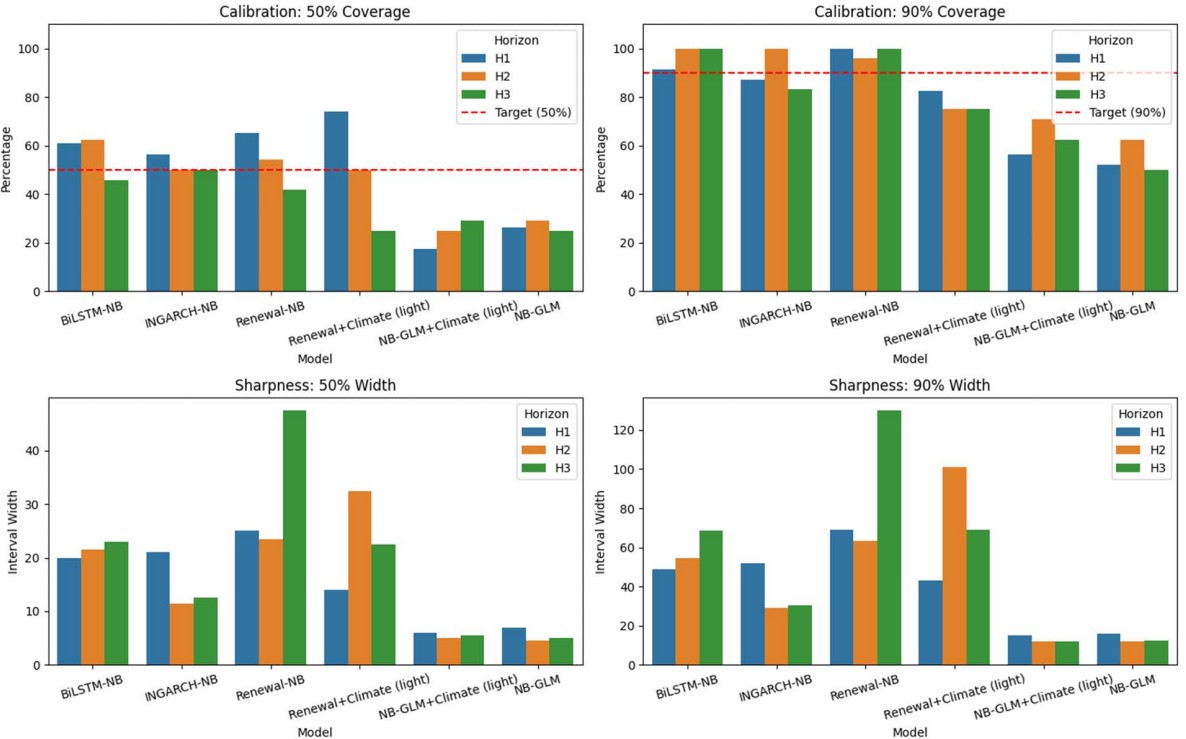

**Fig 13. Era-based calibration and sharpness (2021-2024).** Empirical coverage (50%, 90%) and median predictive-interval widths on the 2021–2024 aligned subset.

decision horizon and the calibration-sharpness trade-off. For short horizons, INGARCH-NB provides strong probabilistic accuracy with moderate interval widths and generally good calibration, making it well-suited for operational triggers that depend on distributional accuracy rather than point forecasts alone. For longer horizons, BiLSTM-NB offers a high 90% reliability but at the cost of wider uncertainty; this may be preferable when the cost of missing high-incidence events outweighs the drawbacks of issuing conservative risk bands. In practice, horizon-specific selection rules or lightweight ensembles can exploit these complementary strengths, but action thresholds (e.g., triggers based on predictive quantiles) should be calibrated to empirical coverage rather than assumed nominal performance.

**Climate, seasonality, and feasibility.** We deliberately restricted exogenous inputs to a leakage-safe "light climate" design lag-1 precipitation, temperature, and humidity, to reflect realistic conditions where real-time climate products can be delayed or inconsistent. Under these constraints, climate augmentation yielded only modest improvements for NB-GLM and did not improve renewal forecasts; Diebold-Mariano (DM) tests did not show statistically significant gains at the 5% level. This does *not* imply that climate is unimportant for dengue transmission. Rather, it suggests that (i) the restricted, lag-only feature set; (ii) the linear structure of the NB-GLM; and (iii) the sensitivity of renewal calibration to specification choices may limit the realized benefit at a monthly resolution. For deployment elsewhere, richer meteorological nowcasts and entomological covariates may improve skill; however, this requires verifying their latency and availability to avoid leakage and explicitly handling missingness and backfilling.

**Calibration, sharpness, and reliability.** A key operational lesson is that coverage alone can be misleading. The NB-GLM baselines produced the narrowest intervals but severely under-covered, indicating overconfidence that may bias

decisions toward under-preparedness. Conversely, renewal-based forecasts sometimes achieve high coverage largely by inflating uncertainty, which degrades the log score and reduces the practical value of forecasts for targeting interventions. INGARCH-NB and BiLSTM-NB occupied a more useful middle ground: INGARCH-NB combined strong log scores with generally adequate calibration and comparatively compact intervals (especially at $h=2$), while BiLSTM-NB emphasized tail reliability at longer horizons. These patterns underscore the importance of monitoring calibration jointly with sharpness (e.g., coverage alongside interval width and PIT diagnostics) when forecasting is used to trigger resource allocation.

**Heterogeneity and outbreak conditions.** Regime-stratified analyses highlight that assessing performance during outbreak months is difficult at a monthly cadence due to the small and highly influential nature of the outbreak subset. Nevertheless, the results illustrate a consistent phenomenon: some models avoid upper-tail misses during outbreaks primarily by issuing very wide predictive intervals, a strategy that does not necessarily indicate superior calibration. During non-outbreak months, INGARCH-NB remained the strongest in mean log score across horizons, while in outbreaks, rankings were more variable and tightly coupled to the accuracy-sharpness trade-off. This motivates a conservative interpretation of outbreak-specific rankings and suggests that prospective use should incorporate safeguards (e.g., horizon-specific uncertainty monitoring and explicit rules for when to trigger heightened preparedness).

**Interpretability versus performance.** Mechanistic renewal models remain appealing because they support epidemiological interpretation through $R_t$-like constructs. However, at a monthly resolution, the assumed serial kernel and aggregation choices can materially affect both mean predictions and uncertainty. Furthermore, misspecification can induce diffuse tails that are heavily penalized by proper scoring rules. A pragmatic compromise for early warning is dual reporting: operational probabilistic forecasts derived from the best-performing statistical or deep learning model, paired with mechanistic summaries (e.g., renewal-based $R_t$ trajectories) for interpretability and situational awareness, treating the latter cautiously when calibration diagnostics indicate instability.

**Comparison with prior work.** The horizon-dependent trade-offs we observe align with broader forecasting evidence: autoregressive count models often excel at shorter leads; flexible sequence models can maintain reliability as horizons extend; and mechanistic models require careful specification and appropriately resolved data to be competitive. Importantly, our evaluation design emphasized aligned issue-target comparisons and leakage controls, reducing the risk of overstating gains from exogenous covariates or complex architectures.

**Strengths and limitations.** Strengths of this study include a unified probabilistic evaluation (using the mean log score as the primary metric), explicit calibration and sharpness diagnostics, leakage-safe handling of climate covariates, and aligned backtests that support fair comparisons. However, limitations are notable. First, results are based on a single city and monthly data; generalizability across settings, reporting practices, and spatial heterogeneity remains to be established. Second, the light climate design may underutilize environmental information, but it reflects a deliberate feasibility constraint. Third, we did not explicitly model reporting delays or structural breaks; operational systems may benefit from adaptive schemes (e.g., change-point detection or robust retraining triggers). Fourth, the outbreak subset in regime stratification is small, so outbreak-specific conclusions should be considered descriptive.

**Guidance for scale-up.** For agencies considering implementation, we suggest: (1) adopting horizon-aware deployment (e.g., INGARCH-NB as a strong default, and BiLSTM-NB when tail reliability at longer horizons is prioritized); (2) synchronizing forecast issuance with operational decision calendars; (3) monitoring calibration online via rolling coverage and PIT dashboards and retraining when deviations persist; (4) setting action thresholds using retrospective empirical coverage to mitigate overconfidence; and (5) auditing climate feed latency and backfill behavior before expanding exogenous inputs. These steps align methodological rigor with institutional capacity and support equitable uptake.

**Future directions.** Three methodological avenues appear promising: (i) hybrid renewal-RNN models that retain mechanistic interpretability while learning residual structure; (ii) probabilistic ensembling across model classes to leverage complementary strengths; and (iii) hierarchical sharing across districts to improve data efficiency and spatial

generalization. Substantively, integrating vector surveillance, mobility proxies, and high-resolution climate nowcasts could enhance predictive skill, provided leakage safeguards and missing-data policies remain central. Finally, prospective evaluations (including silent trials and decision-impact studies) should accompany rollout to verify that forecast use leads to improved outcomes without exacerbating inequities.

In summary, leakage-safe probabilistic dengue forecasts at a monthly cadence can be operationally useful, but model choice should be guided by the decision horizon and the calibration-sharpness trade-off, rather than by accuracy metrics alone. A horizon-aware portfolio that prioritizes strong distributional accuracy at shorter horizons and reliable tails at longer ones offers a practical path for early warning in settings like Freetown, with broader validation and responsible systems integration representing the critical next steps.

## Conclusion

We developed and compared leakage-safe probabilistic dengue forecasting models for Freetown, Sierra Leone (2015–2024) at a monthly cadence. Our study spanned statistical count models (NB–GLM, INGARCH–NB), a mechanistic renewal model (Renewal–NB), and a deep sequence model with a negative binomial output (BiLSTM–NB), evaluated under an expanding-window, rolling-origin design. Using aligned evaluation sets and proper scoring rules, we found that INGARCH-NB achieved the strongest overall distributional accuracy on the global aligned set across horizons $h \in \{1, 2, 3\}$, Meanwhile, BiLSTM-NB remained competitive, delivering particularly reliable 90% predictive interval coverage at longer horizons, albeit with wider intervals. In contrast, NB-GLM variants tended to be overconfident (under-covered), whereas renewal-based specifications attained nominal coverage primarily through uncertainty inflation, which reduced sharpness and penalized log scores. A leakage-safe "light climate" design, incorporating lag-1 precipitation, temperature, and humidity, yielded modest, model-dependent improvements for NB-GLM, though these were not statistically significant at conventional levels and did not improve renewal forecasts.

Operationally, these findings support a horizon-aware forecasting strategy: INGARC-NB serves as a strong default for near- and medium-term planning where distributional accuracy and moderate sharpness are required, while BiLSTM-NB offers a complementary option when conservative tail reliability is prioritized at longer horizons. Study limitations include the monthly temporal resolution, the focus on a single urban setting, and the omission of spatial structure and immunity or serotype dynamics. Future work should evaluate higher-frequency data, multi-city transferability, hierarchical and hybrid (mechanistic-learning) models, and prospective real-time pipelines linked to explicit public health decision triggers. Overall, this study demonstrates that principled probabilistic forecasting with leakage controls and aligned evaluation can provide actionable, uncertainty-aware dengue guidance for public health practice.

## Supporting information

**S1 Fig. PIT histograms by model and horizon (aligned evaluation).** Probability integral transform (PIT) histograms for aligned forecasts by model and horizon. Deviations from uniformity indicate miscalibration (e.g., over- or under-dispersion and systematic bias).
(TIF)

**S2 Fig. Model performance heatmap (aligned evaluation).** Displays mean log scores across models and horizons.
(TIF)

**S3 Fig. Model performance heatmap (Annotations).** Numerical overlays represent empirical 90% predictive-interval coverage for calibrated uncertainty assessment.
(TIF)

**S4 Fig. Regime dashboard at horizon $h = 1$ (aligned evaluation).** Diagnostics stratified by regime (non-outbreak vs outbreak) for 1-month-ahead targets: (i) mean log score (higher/less negative is better), (ii) empirical coverage of nominal

50% and 90% predictive intervals, (iii) median predictive-interval widths (50%, 90%), and (iv) upper-tail miss rate (percentage of targets exceeding the upper 90% PI bound). Outbreak months are defined by $y_{t+1} > \text{thr}_1$ with $\text{thr}_1 = 33.00$. The outbreak subset is small; interpret descriptively and jointly with interval widths (high coverage may reflect diffuse forecasts). (TIF)

**S5 Fig. Regime dashboard at horizon $h = 2$ (aligned evaluation).** Same diagnostics as S4 Fig for 2-month-ahead targets, with outbreaks defined by $y_{t+2} > \text{thr}_2$ and $\text{thr}_2 = 32.50$. Highlights horizon-dependent changes in calibration and sharpness under outbreaks. (TIF)

**S6 Fig. Regime dashboard at horizon $h = 3$ (aligned evaluation).** Same diagnostics as S4 Fig for 3-month-ahead targets, with outbreaks defined by $y_{t+3} > \text{thr}_3$ and $\text{thr}_3 = 32.25$. At longer horizons, sharpness differences can be substantial; near-zero tail-miss rates during outbreaks may coincide with excessively wide intervals. (TIF)

**S1 Table. Hyperparameter settings for all models.** Summary of final model configurations used in the main experiments (e.g., GLM covariates/penalty if any, INGARCH order and link, renewal kernel and seasonal $R_t$ specification, BiLSTM architecture/training settings, and calibration settings). (Provided in a separate upload.). (DOCX)

**S1 Data. De-identified monthly dengue cases and climate aggregates (2015–2024).** Available at: https://doi.org/10.34740/kaggle/dsv/13257213 (CSV)

## Author contributions

**Data curation:** Michael Marko Sesay.

**Software:** Michael Marko Sesay.

**Supervision:** Antony Ngunyi, Herbert Imboga.

**Validation:** Michael Marko Sesay.

**Visualization:** Michael Marko Sesay.

**Writing – original draft:** Antony Ngunyi, Herbert Imboga.

**Writing – review & editing:** Antony Ngunyi, Herbert Imboga.

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
