## [Decision Letter · Decision Letter 0]

20 Jan 2026

PGPH-D-25-03170

Probabilistic Forecasting of Monthly Dengue Cases Using Epidemiological and Climate Signals: A BiLSTM–Naive Bayes Model Versus Mechanistic and Count-Model Baselines.

Dear Dr. Sesay,

Thank you for submitting your manuscript to PLOS Global Public Health. After careful consideration, we feel that it has merit but does not fully meet PLOS Global Public Health’s publication criteria as it currently stands. Therefore, we invite you to submit a revised version of the manuscript that addresses the points raised during the review process.

The manuscript has been evaluated by two reviewers, and their comments are available below.

The reviewers have raised a number of concerns that need attention. In particular, they request additional analyses, improved clarity and justification for the methods, and further discussion.

Could you please revise the manuscript to carefully address the concerns raised?

We look forward to receiving your revised manuscript.

Kind regards,

Helen Howard

Staff Editor

Journal Requirements:

1. We ask that a manuscript source file is provided at Revision. Please upload your manuscript file as a .doc, .docx, .rtf or .tex.

2. Please upload separate figure files in .tif or .eps format. Also, remove the figures from your manuscript file but keep the legends.

3. We have noticed that you have a list of Supporting Information legends in your manuscript. However, there are no S3 File uploaded to the submission. Please upload them as separate files with the item type 'Supporting Information'.

Additional Editor Comments (if provided):

Reviewers' comments:

Reviewer's Responses to Questions

**Comments to the Author**

1. Does this manuscript meet PLOS Global Public Health’s publication criteria ? Is the manuscript technically sound, and do the data support the conclusions? The manuscript must describe methodologically and ethically rigorous research with conclusions that are appropriately drawn based on the data presented.? Is the manuscript technically sound, and do the data support the conclusions? The manuscript must describe methodologically and ethically rigorous research with conclusions that are appropriately drawn based on the data presented.

Reviewer #1: Yes

Reviewer #2: Yes

2. Has the statistical analysis been performed appropriately and rigorously?

Reviewer #1: N/A

Reviewer #2: Yes

3. Have the authors made all data underlying the findings in their manuscript fully available (please refer to the Data Availability Statement at the start of the manuscript PDF file)?

The PLOS Data policy requires authors to make all data underlying the findings described in their manuscript fully available without restriction, with rare exception. The data should be provided as part of the manuscript or its supporting information, or deposited to a public repository. For example, in addition to summary statistics, the data points behind means, medians and variance measures should be available. If there are restrictions on publicly sharing data—e.g. participant privacy or use of data from a third party—those must be specified.requires authors to make all data underlying the findings described in their manuscript fully available without restriction, with rare exception. The data should be provided as part of the manuscript or its supporting information, or deposited to a public repository. For example, in addition to summary statistics, the data points behind means, medians and variance measures should be available. If there are restrictions on publicly sharing data—e.g. participant privacy or use of data from a third party—those must be specified.

Reviewer #1: Yes

Reviewer #2: No

4. Is the manuscript presented in an intelligible fashion and written in standard English?

Reviewer #1: Yes

Reviewer #2: Yes

5. Review Comments to the Author

Reviewer #1: Title: Probabilistic Forecasting of Monthly Dengue Cases Using Epidemiological and Climate Signals: A BiLSTM–Naive Bayes Model Versus Mechanistic and Count-Model Baselines.

Manuscript Number: PGPH-D-25-03170

This manuscript presents a rigorous comparative study of probabilistic forecasting models for monthly dengue incidence in Freetown, Sierra Leone, covering the period 2015–2025. It evaluates four major model classes—NB-GLM, INGARCH-NB, Renewal-NB, and BiLSTM-NB—under a leakage-safe rolling-origin evaluation. The article demonstrates strong methodological maturity, careful control of data leakage, and thorough probabilistic evaluation using proper scoring rules, interval coverage, sharpness metrics, PIT diagnostics, and Diebold–Mariano tests.

The manuscript is generally well-written, technically sound, and addresses an important operational public health problem. It positions itself as one of the few works offering aligned comparisons of mechanistic, statistical, and deep-learning models under realistic constraints for West African dengue surveillance.

This article presents a methodologically rigorous comparison of four probabilistic forecasting approaches—NB-GLM, INGARCH-NB, Renewal-NB, and BiLSTM-NB—applied to monthly dengue case data from Freetown, Sierra Leone (2015–2025). The study addresses an important gap by evaluating mechanistic, statistical, and deep-learning models under aligned, leakage-safe conditions. While the work is comprehensive and technically strong, several critical issues affect its accessibility, interpretability, and broader applicability.

Strengths

The study excels in methodological rigor. Its strict leakage safeguards, careful feature-timing rules, and use of expanding-window rolling-origin evaluation significantly strengthen reliability. The inclusion of proper scoring rules, interval coverage, sharpness metrics, PIT histograms, and Diebold–Mariano tests provides a complete probabilistic evaluation rarely seen in dengue forecasting studies. The horizon-specific findings—INGARCH-NB outperforming at 1–2 months and BiLSTM-NB excelling at 3 months—are well supported by aligned comparisons and statistical significance tests. The transparency of data, code, and alignment artefacts enhances reproducibility and credibility. Additionally, the manuscript offers practical guidance for operational forecasting, including a realistic “light climate” input strategy suitable for resource-limited settings.

Limitations

Despite its strengths, the manuscript is heavily technical, with extensive mathematical exposition in the main text. This may limit accessibility for public-health practitioners who are likely part of the target audience. The mechanistic renewal model is presented as a baseline but is arguably underspecified; the use of a short, fixed 3-month kernel may not realistically capture dengue’s generation interval dynamics, likely contributing to its poor performance. This limits the interpretive value of the mechanistic comparison. This limitation should be addressed.

The study’s climate treatment, while intentionally conservative, may underexploit important environmental drivers. Although justified operationally, this constraint restricts exploration of potentially meaningful lag structures or seasonal climate anomalies. The analysis is limited to a single city and monthly data frequency, raising questions about generalizability across geographies with different climate patterns and dengue transmission dynamics. Moreover, the monthly temporal resolution may obscure rapid outbreak shifts, possibly disadvantaging mechanistic and hybrid models that rely on finer-grained dynamics. This should be addressed.

The manuscript makes a valuable and original contribution to dengue forecasting, offering robust methodological innovations and practical insights for real-time surveillance systems. However, improved clarity, stronger justification for mechanistic assumptions, and expanded discussion of generalizability would enhance its usefulness and scholarly impact. With revisions to improve accessibility and contextual depth, the study is well positioned for publication and for informing operational forecasting practice in similar settings.

Reviewer #2: 1. What is PIT in the abstract stand for? The authors should avoid using abbreviations in the abstract.

2. The authors should providing some additional analysis, such as experimenting with alternative or longer serial-interval kernels, or simple sensitivity checks (e.g., different window lengths, or, if possible, finer temporal resolution).

3. Please, justifies the small climate feature set, mentioning any exploratory work with larger sets.

4. The authors should add a clearly labelled missing-data handling subsection that specifies: The imputation method, the number of imputed months, and how they were used in training/evaluation, plus any sensitivity.

5. While the architecture, optimization, and calibration steps are described, the process for choosing hyperparameters is not fully audit-ready.

6. I recommend that the authors conduct an additional experiment to demonstrate the generalizability of the proposed model.

6. PLOS authors have the option to publish the peer review history of their article (what does this mean? ). If published, this will include your full peer review and any attached files.). If published, this will include your full peer review and any attached files.

**Do you want your identity to be public for this peer review?** For information about this choice, including consent withdrawal, please see our Privacy Policy ..

Reviewer #1: **Yes:** Shyam Sanjeewa Nishantha PereraShyam Sanjeewa Nishantha Perera

Reviewer #2: No

Figure Resubmissions:

---

## [Decision Letter · Decision Letter 1]

2 Mar 2026

Probabilistic Forecasting of Monthly Dengue Cases Using Epidemiological and Climate Signals: A BiLSTM-Negative Binomial Model Versus Mechanistic and Count-Model Baselines

PGPH-D-25-03170R1

Dear Mr Sesay,

We are pleased to inform you that your manuscript 'Probabilistic Forecasting of Monthly Dengue Cases Using Epidemiological and Climate Signals: A BiLSTM-Negative Binomial Model Versus Mechanistic and Count-Model Baselines' has been provisionally accepted for publication in PLOS Global Public Health.

Best regards,

Julia Robinson

Executive Editor

Reviewer Comments (if any, and for reference):

Reviewer's Responses to Questions

**Comments to the Author**

1. If the authors have adequately addressed your comments raised in a previous round of review and you feel that this manuscript is now acceptable for publication, you may indicate that here to bypass the “Comments to the Author” section, enter your conflict of interest statement in the “Confidential to Editor” section, and submit your "Accept" recommendation.

Reviewer #1: All comments have been addressed

2. Does this manuscript meet PLOS Global Public Health’s publication criteria ? Is the manuscript technically sound, and do the data support the conclusions? The manuscript must describe methodologically and ethically rigorous research with conclusions that are appropriately drawn based on the data presented.? Is the manuscript technically sound, and do the data support the conclusions? The manuscript must describe methodologically and ethically rigorous research with conclusions that are appropriately drawn based on the data presented.

Reviewer #1: Yes

3. Has the statistical analysis been performed appropriately and rigorously?

Reviewer #1: Yes

4. Have the authors made all data underlying the findings in their manuscript fully available (please refer to the Data Availability Statement at the start of the manuscript PDF file)?

The PLOS Data policy requires authors to make all data underlying the findings described in their manuscript fully available without restriction, with rare exception. The data should be provided as part of the manuscript or its supporting information, or deposited to a public repository. For example, in addition to summary statistics, the data points behind means, medians and variance measures should be available. If there are restrictions on publicly sharing data—e.g. participant privacy or use of data from a third party—those must be specified.requires authors to make all data underlying the findings described in their manuscript fully available without restriction, with rare exception. The data should be provided as part of the manuscript or its supporting information, or deposited to a public repository. For example, in addition to summary statistics, the data points behind means, medians and variance measures should be available. If there are restrictions on publicly sharing data—e.g. participant privacy or use of data from a third party—those must be specified.

Reviewer #1: Yes

5. Is the manuscript presented in an intelligible fashion and written in standard English?

Reviewer #1: No

6. Review Comments to the Author

Reviewer #1: NA

7. PLOS authors have the option to publish the peer review history of their article (what does this mean? ). If published, this will include your full peer review and any attached files.). If published, this will include your full peer review and any attached files.

**Do you want your identity to be public for this peer review?** For information about this choice, including consent withdrawal, please see our Privacy Policy ..

Reviewer #1: No
